# Clinical validation of engineered CRISPR/Cas12a for rapid SARS-CoV-2 detection

Long T. Nguyen[1], Santosh R. Rananaware[1], Brianna L. M. Pizzano [2], Brandon T. Stone[3] & Piyush K. Jain [1,4✉]

## Abstract

**Background** The coronavirus disease (COVID-19) caused by SARS-CoV-2 has swept through the globe at an unprecedented rate. CRISPR-based detection technologies have emerged as a rapid and affordable platform that can shape the future of diagnostics.

**Methods** We developed ENHANCEv2 that is composed of a chimeric guide RNA, a modified LbCas12a enzyme, and a dual reporter construct to improve the previously reported ENHANCE system. We validated both ENHANCE and ENHANCEv2 using 62 nasopharyngeal swabs and compared the results to RT-qPCR. We created a lyophilized version of ENHANCEv2 and characterized its detection capability and stability.

**Results** Here we demonstrate that when coupled with an RT-LAMP step, ENHANCE detects COVID-19 samples down to a few copies with 95% accuracy while maintaining a high specificity towards various isolates of SARS-CoV-2 against 31 highly similar and common respiratory pathogens. ENHANCE works robustly in a wide range of magnesium concentrations (3 mM-13 mM), allowing for further assay optimization. Our clinical validation results for both ENHANCE and ENHANCEv2 show 60/62 (96.7%) sample agreement with RT-qPCR results while only using 5 µL of sample and 20 minutes of CRISPR reaction. We show that the lateral flow assay using paper-based strips displays 100% agreement with the fluorescence-based reporter assay during clinical validation. Finally, we demonstrate that a lyophilized version of ENHANCEv2 shows high sensitivity and specificity for SARS-CoV-2 detection while reducing the CRISPR reaction time to as low as 3 minutes while maintaining its detection capability for several weeks upon storage at room temperature.

**Conclusions** CRISPR-based diagnostic platforms offer many advantages as compared to conventional qPCR-based detection methods. Our work here provides clinical validation of ENHANCE and its improved form ENHANCEv2 for the detection of COVID-19.

## Plain language summary

The COVID-19 pandemic has underscored the need for rapid and accurate tests to detect SARS-CoV-2 infection. The tests commonly used have limitations, and a detection system based on CRISPR technology could offer a useful alternative. CRISPR is a technology derived from bacteria that can specifically detect pieces of DNA. We have previously developed ENHANCE, a detection system that converts the SARS-CoV-2 genetic material into DNA that is then detected by an engineered CRISPR technology. Here, we develop an improved version of this method, ENHANCEv2, that has an extended shelf life and less need for refrigeration, facilitating transportation of the components required for the test and its use. We show that both ENHANCE and ENHACEv2 can quickly and accurately detect SARS-CoV-2 in swabs from infected people. This is a step towards having more versatile tools to detect SARS-CoV-2 infection quickly and accurately.

[1] Department of Chemical Engineering, University of Florida, Gainesville, FL, USA. [2] Department of Agricultural and Biological Engineering, University of Florida, Gainesville, FL, USA. [3] Department of Microbiology, University of Florida, Gainesville, FL, USA. [4] UF Health Cancer Center, University of Florida, Gainesville, FL, USA. ✉email: Jainp@ufl.edu

With a global pandemic of over 72 million COVID-19 cases resulting in over 1.6 million deaths, there remains a crucial need for diagnostic tools that allow for quick yet accurate detection of SARS-CoV-2 without the requirement of expensive equipment and extensive training[1,2]. While several vaccines are under Phase III clinical trials or have been granted Emergency Use Authorizations (EUAs) by the FDA[3–7], several variants of concern are emerging. The second, third, and fourth waves have impacted many countries across the globe, and with treatments still somewhat limited, improvements in testing are more necessary than ever to keep both case numbers and fatality numbers down in order for preventative measures to be effective down the line[8,9].

SARS-CoV-2 is a ~30 kb betacoronavirus composed of four known structural proteins, including the spike (S), nucleocapsid (N), membrane (M), and envelope (E) proteins, as well as a viral RNA genome[10]. Nucleic acid detection methods often target N, E, and RdRp (RNA-dependent RNA polymerase) genes. In quantitative Reverse Transcription Polymerase Chain Reaction (RT-qPCR), the viral RNA is reverse transcribed into complementary DNA (cDNA) and then amplified through cyclic changing of temperatures to achieve denaturation of nucleic acid strands, annealing of primers to the template DNA, and extension of new complementary DNA by the addition of nucleotides by a polymerase. DNA copies are detected in real-time through the emission of fluorescence-based reporters. Detection using this method requires expensive equipment and clinically trained professionals, and is not easily portable[11,12].

CRISPR-Cas-based methods such as SHERLOCK (Specific High sensitivity Enzymatic Reporter unLOCKing) and DETECTR (DNA Endonuclease-Targeted CRISPR Trans Reporter) have recently received an EUA by the Food and Drug Administration (FDA). They take advantage of the collateral cleavage (trans) activity from Class 2 Type V and Type VI Cas proteins, specifically Cas13a and Cas12a, to cleave a FRET-based reporter resulting in fluorescence. Cas12a-based DETECTR has a limit of detection of around 20 copies/μL, while SHERLOCK is based on Cas13a and is time-consuming (~1 h, as opposed to ~30 min) but can detect ~ 6.75 copies/μL (Supplementary Table S1)[13–22]. To amplify the target genes while accommodating the drawbacks of traditional PCR, both COVID-19 detection platforms pair CRISPR/Cas reactions with an antecedent isothermal amplification step, with SHERLOCK utilizing Recombinase Polymerase Amplification (RPA) and DETECTR utilizing Reverse Transcription Loop-Mediated Isothermal Amplification (RT-LAMP). The RT-LAMP, performed at a constant temperature of 60-65 °C, is less time-consuming due to the incorporation of loop primers that facilitate subsequent primer binding and amplification at multiple sites. It is also more sensitive than RPA and can be easily adopted without supply-chain issues[23].

Here we report the clinical validation of the ENHANCE system, which was developed in our previous study[24] and it utilizes a 7-nucleotide DNA extension on the 3' end of the crRNA to boost the collateral cleavage activity of a LbCas12a protein, enabling an increase in sensitivity while maintaining specificity even at low magnesium concentrations. Similar to DETECTR, RT-LAMP is used for a pre-amplification step prior to the CRISPR-Cas reaction, which trans-cleaves the fluorophore from the quencher after the initial cis-cleavage of the target dsDNA. In order to validate the ENHANCEv1 system under clinical conditions, 62 nasopharyngeal samples (31 positive samples and 31 negative samples) were tested using both the fluorescence-based and lateral flow assays and then compared to RT-qPCR results. Further, we developed a lyophilized version of the ENHANCEv2 system, which utilizes modified crRNAs from ENHANCE, a mutated LbCas12a protein (Cas12a$^{D156R}$) to further amplify the signal[25–28], and a dual reporter construct that allows each sample to be read using a fluorescence-based and a lateral flow assay format in the same reaction. The lyophilized CRISPR reaction in ENHANCEv2 occurs at an accelerated rate and preserves for a long period of time upon room temperature storage.

## Methods

**Protein expression and purification**. LbCas12a gene fragment was obtained by PCR from the plasmid LbCpf1-2NLS as a gift from Jennifer Doudna (Addgene plasmid # 102566)[29] and subcloned into a linearized CL7-tagged vector (Plasmid #21, TriAltus Bioscience) by digesting with HindIII and XhoI. The fragments assembly was performed using NEBHiFi Builder (New England Biolabs) following the manufacturer's protocol. The assembled plasmid was transformed into DH5α (New England Biolabs) competent cells. Individual colonies were picked the next day and inoculated in 10 ml of LB broth, Miller (Fisher Scientific) at 37 °C overnight. Cells were harvested, and plasmids were extracted and purified using the Monarch Mini Plasmid prep (New England Biolabs). The CL7-tagged LbCas12a and LbCas12a$^{D156R}$ expression plasmids are made available on Addgene (#164658 and 164659, respectively).

For protein expression, 100 ng of the purified plasmid was transformed into BL21(DE3) competent cells (New England Biolabs). Individual colonies were picked and inoculated in 10 ml of Terrific Broth (TB) at 37 °C for 8–10 h. The culture was then added to 1 L TB broth containing 50 μg/ml Kanamycin (Fisher Scientific) and 50 μL antifoam 204 (Sigma Aldrich) and let grown until OD 600 = 0.6. The culture was then taken out of the 37 °C incubator and let cooled on ice for 30–45 min. Next, 0.5 mL of 1M isopropyl β-D-1-thiogalactopyranoside (IPTG, Fisher Scientific) was added to the culture and let grown at 18 °C overnight.

The overnight culture was centrifuged to collect cell pellets the next day. They were resuspended in Lysis Buffer A (2 M NaCl, 50 mM Tris-HCl, pH = 7.5, 0.5 mM TCEP, 5% Glycerol, 1 mM PMSF, 0.25 mg/ml lysozyme). The mixture was disrupted by sonication, centrifuged at 40,000×g, and filtered through a 0.45 μm filter. The lysate was run through a 1 ml CL7/Im7 column (TriAltus Bioscience) connected to the FPLC Biologic Duoflow system (Bio-Rad). The column was washed for at least three cycles of alternating Wash Buffer B (2 M NaCl, 50 mM Tris-HCl, pH = 7.5, 0.5 mM TCEP, 5% Glycerol) and Wash Buffer C (50 mM Tris-HCl, pH = 7.5, 0.5 mM TCEP, 5% Glycerol). The column was then eluted by adding 5 mL of SUMO protease (purified from plasmid pCDB302 as a gift from Christopher Bahl, Addgene plasmid# 113673)[30] and flown through in a closed-loop cycle at 30 °C for 1.5 h. Optionally, to remove SUMO protease, the eluted solution was then concentrated using a 30 kDa MWCO Sartorius Vivaspin Concentrator to 500 μL and subjected to size exclusion chromatography in SEC buffer (500 mM NaCl, 50 mM Tris-HCl, pH = 7.5, 0.5 mM TCEP) via the Superdex 200 increase 10/300 GL column (Cytiva). Eluted fractions were collected, pooled together, concentrated, quantified using the NanoDrop (Thermo Fisher), snap-frozen in dry ice, and stored at −80 °C until use.

For LbCas12a$^{D156R}$ expression and purification, the CL7-tagged plasmid obtained by subcloning above was mutated using the Q5® Site-Directed Mutagenesis Kit (New England Biolabs) following the manufacturer's protocol. The protein expression and purification were the same as described above.

Bst-LF polymerase expression plasmid was obtained as a gift from Drew Endy & Philippa Marrack (Addgene plasmid # 153313). Br512 (an engineered version of Bst polymerase) and reverse transcriptase RTx (exo-) were obtained as a gift from Andrew Ellington (Addgene plasmid # 161875 and # 145028,

respectively). Bst-LF polymerase and Br512 polymerase were expressed and purified following Maranhao et al.[31], and RTx(exo-) was expressed and purified following Bhadra et al.[32].

**Lyophilization of ENHANCEv2 CRISPR reaction**. To assemble a CRISPR reaction, 100 nM LbCas12a$^{D156R}$, 125 nM crRNA-Mod, 500 nM dual reporter were combined in 1x NEBuffer 2.1 (New England Biolabs) to a total of 50 μL. The mixture was scaled up accordingly to make 5x and 20x reaction aliquots. These aliquots were then subjected to lyophilization using the Labconco freeze dryer for 2–4 days.

**Lyophilization of RT-LAMP reagents**. One reaction of the RT-LAMP assay reagent mixture was prepared by combining 35 nanomoles dNTPs, 2.5 μL of 10X LAMP primer mix, 25 picomoles of Br512 (or Bst-LF) polymerase, 0.1 μg of RTx(exo-), and 1.25 μmoles of D-( + )-trehalose, anhydrous. The mixture was frozen for 2 h at −80 °C prior to freeze-drying using the Labconco lyophilizer for 24 h.

The lyophilized reaction mixture was reconstituted with 1X isothermal buffer (20 mM Tris-HCl, 10 mM (NH$_4$)$_2$SO$_4$, 50 mM KCl, 8 mM MgSO$_4$, 0.4 M Betaine, PH = 8.8 at 25 °C). The RT-LAMP reaction was then readily initiated by adding RNA samples.

**CDC RT-qPCR assay**. The samples were re-validated with real-time RT-qPCR. The reactions were performed using the CDC-recommended Quantabio qScript XLT One-Step RT-qPCR ToughMix (Catalog# 95132-100) and TaqMan probes and primer sets and measured using the ViiA 7 Real-Time PCR System. RT-qPCR quantification was performed using amplification plots generated by the ViiA 7 software.

**Viral nucleic acid extraction**. For crRNA screening and optimization, LoD estimation, inclusivity testing, and specificity testing, viral RNA extraction was performed using the Lucigen QuickExtract™ DNA Extraction Solution (Cat # QE09050). Viral samples were diluted with QuickExtract™ in a 1:1 (v/v) ratio and incubated at 65 °C for 15 min and 98 °C for 2 min.

For clinical validation experiments, all patient samples were extracted using Maxwell® RSC 16 automated nucleic acid extraction instrument. Maxwell® RSC Viral Total Nucleic Acid Purification Kit (Cat# AS1330, as recommended by CDC) was used for all extractions following the manufacturer's protocol.

**RT-LAMP reactions**. A set of six LAMP primers were designed for each gene using the freely available PrimerExplorer software (https://primerexplorer.jp/e/)[33]. The designed primers were synthesized by Integrated DNA Technologies. A 10x primer mix for each gene was created by mixing the six primers to a final concentration of 16 μM (FIP/BIP), 8 μM (LB/LF), and 2 μM (F3/B3). RT-LAMP master mix (including the positive and negative control) were prepared by combining the WarmStart® Colorimetric LAMP 2X Master Mix with UDG (New England Biolabs) and 10X LAMP primer mixes to a 1X final concentration and total volume of 20 μL. The RT-LAMP reaction was initiated by the addition of target RNA and incubated at 65 °C for 30 min to allow for adequate amplification. The amplified products were tested downstream using the CRISPR-ENHANCE assays.

**Screening of crRNAs and optimization of ENHANCE for detection of SARS-CoV-2**. Genomic RNA from SARS-CoV-2, Isolate USA-WA1/2020 (NR-52285) obtained from Biodefense and Emerging Infections Research Resources Repository (BEI resources), was spiked in the nucleic acid extract obtained from a healthy donor. The spiked extract was amplified for the target genes (N1, N2, E1, E2, R1, and R2) using RT-LAMP. The amplified products were then detected using wild-type CRISPR/Cas12a as well as CRISPR ENHANCEv1.

**Inclusivity testing**. SARS-CoV-2 Genomic RNA of isolates obtained from different geographic regions such as Italy, Hong Kong, and the USA (NR-52498, NR-52388, and NR-52285) obtained from BEI resources were spiked in the nucleic acid extract obtained from the nasopharyngeal swab of a healthy donor. Target genes were amplified using the RT-LAMP protocol described above and detected using CRISPR ENHANCEv1.

**Specificity testing**. To demonstrate the specificity of our assay towards SARS-CoV-2 we obtained 31 highly similar and commonly circulating pathogens from ZeptoMetrix (Cat# NATPPQ-BIO, Cat# NATRVP-3, Cat# NATPPA-BIO). Each pathogen was spiked in a matrix composed of a nasopharyngeal swab from a healthy donor. The spiked nasal swab was extracted using Lucigen QuickExtract™ and target genes within the extracted nucleic acids were amplified using RT-LAMP. The amplified products were detected using CRISPR ENHANCEv1.

**Estimation of LoD**. A nasopharyngeal swab sample from a healthy donor was mixed with an equal volume of Lucigen QuickExtract™ DNA Extraction Solution (Cat # QE09050) and incubated at 65 °C for 15 min and 98 °C for 2 min to extract nucleic acids from the swab. Mock clinical patient samples were prepared by serially diluting the nucleic acid extract with Quantitative PCR (qPCR) Control RNA from Heat-Inactivated SARS-Related Coronavirus 2 (BEI NR-52347) to a final concentration range of 200 copies/μL to 0.2 copies/μL. The LoD for each gene was determined by amplifying the gene using RT-LAMP and then detecting it at the indicated concentrations with CRISPR ENHANCEv1. The LoDs for the N2 gene and E2 gene were confirmed by testing with 20 replicates at 1x and 2x of the previously estimated LoD for each gene.

**Fluorescence-based reporter detection assay**. All fluorescence-based detection experiments were performed in a 384-well plate. CRISPR/Cas complexation was carried out by combining 30 nM LbCas12a and 60 nM crRNA in 1X NEBuffer 2.1 and incubating at 37 °C for 15 min before transferring to a 384-well plate containing 500 nM Fluorophore-Quencher (FQ) and 2 μL of RT-LAMP product. Emitted fluorescence resulting from Cas12a-based trans-cleavage was measured using BioTek Synergy 2 microplate reader with fluorescence measurement at excitation and emission wavelengths of 485/20 and 528/20, respectively, every 2.5 min. For a 96-well plate format, all reagents are scaled up 2.5 times.

**Lateral flow detection assay**. The detection reaction for lateral flow assay was carried out by combining 30 nM LbCas12a, 60 nM crRNA, 200 nM FAM-Biotin reporter in 1X NEBuffer 2.1 to a total volume of 48 μL. Two microliters of the corresponding RT-LAMP product was added to the above mixture and incubated at 37 °C for 20 min. A HybriDetect 1 lateral flow strip (Milenia Biotech) was then dipped in the reaction tube and the presence or absence of the target gene was determined based on a visual readout after 2 min. Lateral flow band signals were later quantified by ImageJ.

**Clinical validation of patient samples using CRISPR-ENHANCE**. A pool of 62 patient samples underwent a blind test. Random samples were selected from the pool, and nucleic

acids extracted from those patient samples were subjected to RT-LAMP-based amplification of the N2 gene, E2 gene, and RNASE P gene. The same amplified RT-LAMP products were then detected using both the fluorescence-based reporter detection assay and the lateral flow assay with ENHANCEv1 and with lyophilized ENHANCEv2.

**Ethical statement.** This study was performed under the University of Florida (UF) Institutional Review Board (IRB) protocol IRB202000781, which was approved as a non-human study, and all relevant ethical regulations were followed. De-identified human samples were obtained from the UF Clinical and Translational Science Institute (CTSI) Biorepository, collected under the UF IRB approved protocol IRB20200879, and from a commercial vendor, Boca Biolistics, procured under the IIRB delinking protocol SOP 10-00114 Rev E.

The CTSI Biorepository was approved to collect specimens without informed consent due to the COVID-19 pandemic being an unprecedented public health emergency and it would limit the research if all samples were not included. There was also the option of obtaining informed consent wherever possible. There were specific limits in the amount and type of data allowed to be gathered for those samples collected without informed consent. Some samples being tested for COVID-19 by the UF Pathology Lab came from patients in outlying clinics or hospitals. Although PHI was collected with the samples, no identifiable data or tissue was nor will be subsequently dispensed. All connections of tissue with data have been and will be conducted by honest brokers.

Boca Biolistics (BBL) is an FDA-recommended provider of SARS-CoV-2 biospecimens for research and diagnostic development. BBL provides remnant SARS-CoV-2 swab specimens as remnant (leftover) samples procured from their network of CAP/CLIA accredited partner laboratories across the United States all of whom have been instrumental in providing COVID-19 screening throughout the pandemic. Under Boca's IIRB Delinking protocol samples are procured and de-linked so that no information can be traced back to the individual patient, providing sound and secure de-identification protecting patient identity. Boca's SOP is consistent with the FDA's "Guidance on informed consent for in vitro diagnostic device studies using leftover human specimens that are not individually identifiable". This allows BBL to provide tens of thousands of highly needed SARS-CoV-2 swab specimens that have been instrumental in both the development and validation of diagnostic instruments throughout the world to test for COVID-19.

**Reporting summary.** Further information on research design is available in the Nature Research Reporting Summary linked to this article.

## Results

**CRISPR-ENHANCE demonstrates robust detection across SARS-CoV-2 genes.** Our previous study observed that a 3′-end chimeric DNA-extended crRNA enhanced the rate of trans-cleavage activity (Kcat/Km) of LbCas12a by 3.2-fold and thus facilitated higher sensitivity in nucleic acid detection (Fig. 1a), also known as ENHANCE (Enhanced Analysis of Nucleic acids with CrRNA Extensions). This universal 7-mer DNA extension is spacer-independent and has been tested in various nucleic acid targets such as GFP, HCV, HIV, and SARS-CoV-2[24]. To detect SARS-CoV-2 at a low copy number, both wild-type and ENHANCE require a pre-amplification step such as Reverse Transcription Loop-mediated Isothermal Amplification (RT-LAMP), which reverse transcribes SARS-CoV-2 genomic RNA to complementary cDNA and converts it into dsDNA targets[14,18,34,35].

In this study, selected regions of the SARS-CoV-2 N gene, E gene, and RdRp gene were targeted with pairs of crRNAs (crN, crE, and crR referred to as crRNA targeting N gene, E gene, and R gene, respectively). These designed crRNAs bear a spacer region of greater than 50% GC content (Fig. 1b and Supplementary Table S2). In addition to previously designed crN2, crE1, and crE2 by Broughton et al.[14], we employed three additional crRNAs (crN1, crR1, and crR2) to not only determine optimal crRNAs but also examine the modified LbCas12a trans-cleavage activity compared to the wild-type system. Consistent with our previous study, when targeting SARS-CoV-2 genomic RNA, the modified crRNAs (referred to as crRNA-Mod) exhibited enhanced trans-cleavage activity, up to fivefold, across all targets compared to the wild-type crRNA (crRNA-WT), indicating considerably higher fluorescence signals via the fluorophore-quencher-based reporter assay results (Fig. 1c–e). Improved activity and specificity remained consistent while detecting multiple geographically distributed isolates (Hong Kong, USA, and Italy) (Fig. 1f). The ENHANCE system works robustly with a wide range of magnesium concentrations (3–13 mM), especially at low magnesium concentrations. Thus, ENHANCE enables a potential strategy for screening a library of guide RNAs under various reaction conditions[36].

Previous studies have shown that the variant LbCas12a$^{D156R}$ exhibited a substantially higher gene editing efficiency compared to the wild-type LbCas12a[25–28]. We hypothesized that this mutated LbCas12a$^{D156R}$ would likely increase the trans-cleavage activity due to its correlation between cis-cleavage and trans-cleavage capabilities[24,37]. Having previously modified crRNA for enhanced LbCas12a trans-cleavage activity, we sought to investigate if we could utilize LbCas12a$^{D156R}$ to further increase the sensitivity for nucleic acid detection purposes. As a result, the combination of LbCas12a$^{D156R}$ and 7-mer DNA-extended crRNA showed a markedly higher fluorescence sensitivity than the wild-type LbCas12a-crRNA and a slight enhancement compared to the combination of wild-type LbCas12a and modified crRNA (Fig. 1g). This observation later became the basis for the development of ENHANCEv2.

**CRISPR-ENHANCE SARS-CoV-2-targeted guide RNA optimization.** To seek optimal crRNAs for the ENHANCE platform, we next evaluated the limit of detection (LoD) for the six selected modified crRNAs. Quantitative qPCR SARS-CoV-2 genomic RNA were utilized to make a serial dilution theoretically ranging from 200 copies/μL to 0.2 copies/μL followed by an isothermal pre-amplification RT-LAMP step (1000 copies/rxn to 1 copy/rxn). An estimated LoD was determined based on the lowest copies/μL having 2/3 of replicates showing at least fivefold in fluorescence signal compared to that of the no-template control (NTC) within 20 min. Out of the six modified crRNAs, crN2 and crE2 designed by Broughton et. al. exhibited the lowest estimated limit of detection with 2/3 replicates detected positive at 0.6 copies/μL and 7.9 copies/μL, respectively (Fig. 2a–d). Hence, we selected crN2-Mod and crE2-Mod as the optimal guide RNAs for our clinical validation. The LoDs of crN2-Mod and crE2-Mod were then confirmed by testing 20 replicates ranging from 1X–33X times the estimated LoD. The LoD was established by observing 95% (19/20 replicates) of the target samples being positive. The replicates of both crN2-Mod and crE2-Mod confirmed the LoD to be 15 copies/μL and 25 copies/μL, respectively, within 20 min of CRISPR reaction (Fig. 2e–h).

We noticed a difference in LoD between the pre-amplification reaction with the incorporation of Uracil DNA glycosylase (UDG) and one without UDG. The above reported LoD for N2 and E2 genes were determined using UDG, which seemed to be

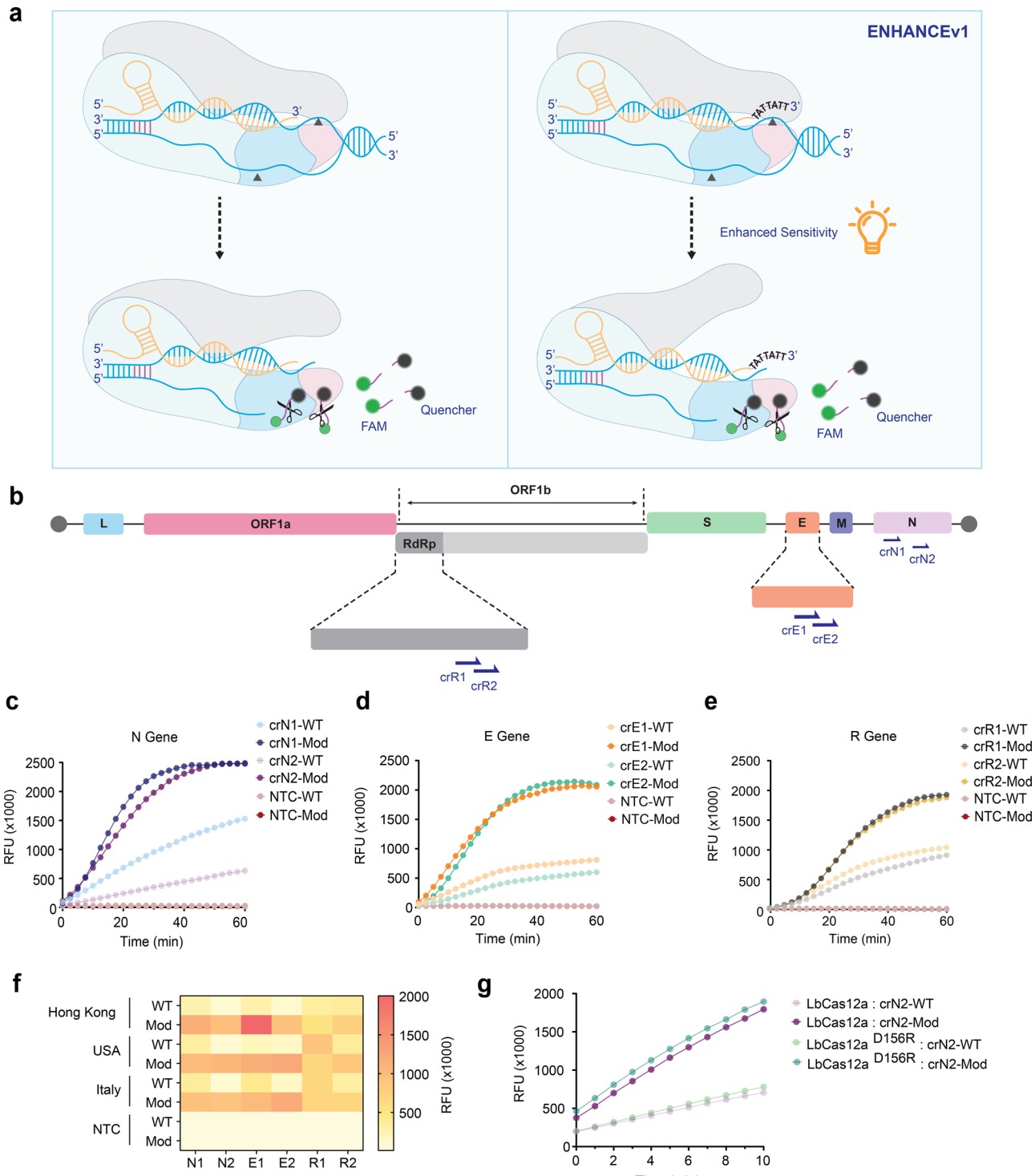

**Fig. 1 crRNA screening and optimization of ENHANCE for detection of SARS-CoV-2. a** Schematic of detection platform using CRISPR-Cas12a highlighting the differences between wild-type and ENHANCE systems. **b** Schematic of SARS-CoV-2 RNA genome and target regions of crRNA used for ENHANCE within the SARS-CoV-2 genomic RNA. **c**–**e** Average fluorescence intensity in RFU of wild-type crRNA (crRNA-WT) and modified crRNA (crRNA-Mod) observed by fluorescence-based reporter assay for N gene, E gene, and RdRp gene, respectively ($n = 6$). **f** SARS-CoV-2 Genomic RNAs obtained from different geographic regions were tested with CRISPR-Cas12a systems ($n = 2$). **g** Comparison of fluorescence intensities when the mutated LbCas12a$^{D156R}$ is introduced ($n = 6$). All targeted sequences are genomic RNA obtained from BEI Resources. Genomic RNAs were diluted down to a concentration of around 1500 copies/μL in Tris-EDTA (TE) buffer followed by 30-min RT-LAMP. Two microliters of the RT-LAMP product was used for the CRISPR detection assay. All fluorescence intensities shown were taken at $t = 20$ min.

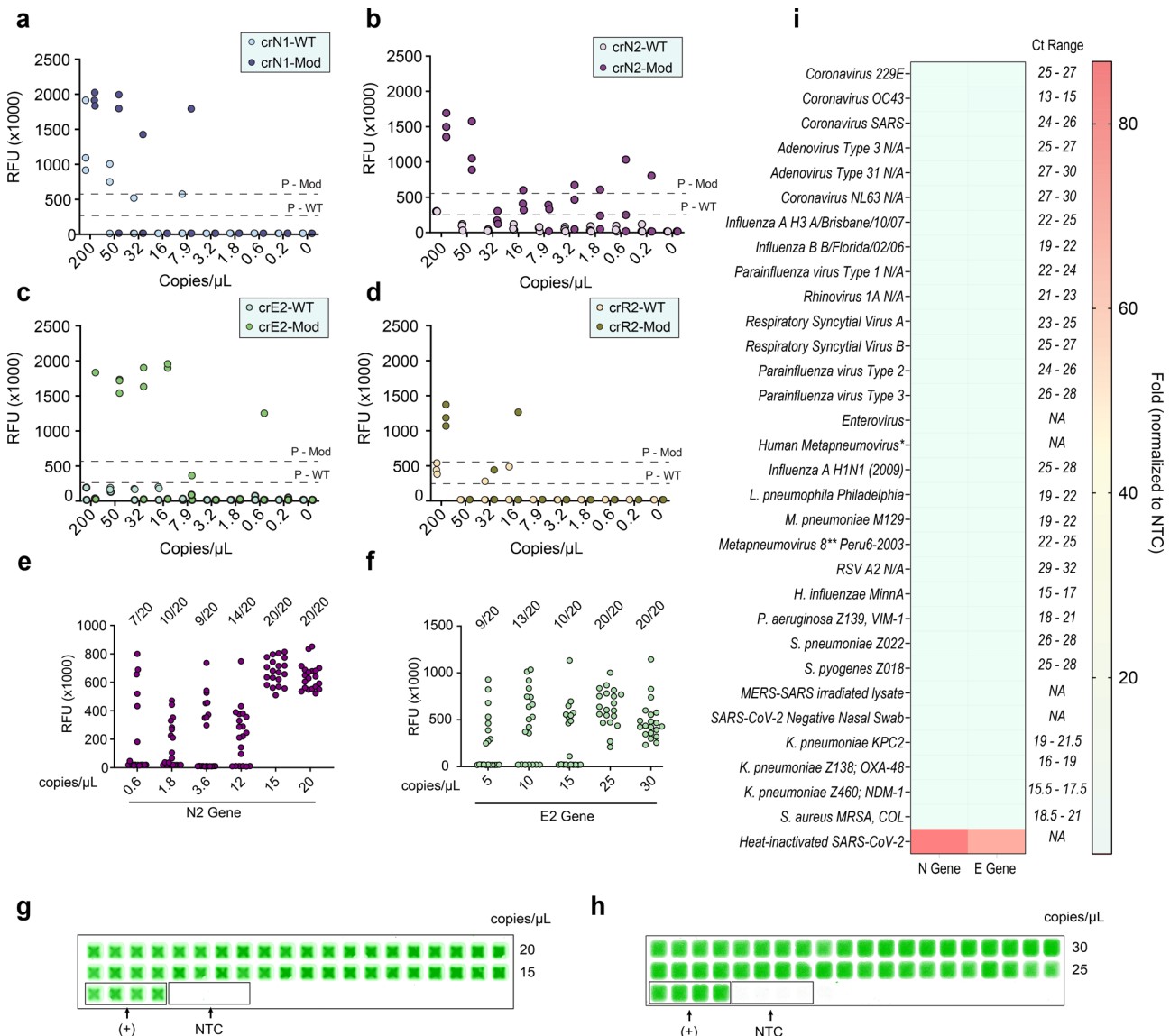

**Fig. 2 Limit of detection and exclusivity testing of ENHANCE. a–d** Fluorescence intensities in RFU of wild-type crRNA (crRNA-WT) and modified crRNA (crRNA-Mod) in serially diluted samples containing quantitative genomic RNA of SARS-CoV-2 ($n = 3$). The RT-LAMP amplification was performed with the addition of UDG since its high sensitivity easily leads to carryover contamination. **e–f** Two out of three of samples detected in **a–d** were subjected to be tested with 20 different serial dilutions around the estimated LoD for N gene and E gene. **g**, **h** Scanned images of samples in **e** and **f**, respectively. **i** Exclusivity testing of ENHANCE against highly similar and commonly circulating pathogens. Genomic RNAs were diluted down to a concentration of around 1500 copies/μL in TE buffer followed by 30-min RT-LAMP. Two microliters of the RT-LAMP product was used for the CRISPR detection assay. Fluorescence signals were taken at $t = 20$ min. All samples with Ct value ranges were obtained from Zeptometrix.

higher than the LoD for the same two genes in the absence of UDG. We detected 19/20 replicates to be positive at 0.2 copies/μL and 7.9 copies/μL, respectively in a 5 μL of sample input volume (Supplementary Fig. S1). As a result, the LoD for N2 violated the Poisson distribution possibly due to carryover contamination.

As a part of exclusivity, we tested a variety of highly similar pathogens to SARS-CoV-2, including the other human coronaviruses (SARS-CoV, MERS-CoV, Coronavirus 229E, OC43, and NL63) and respiratory pathogens commonly detected in nasal swabs or saliva. Notably, the CRISPR-ENHANCE exhibited high specificity towards SARS-CoV-2 with the modified crRNAs targeting the N gene and E gene (Fig. 2i). No cross-reactivity was observed in any of the 31 tested pathogens.

**Validation of ENHANCEv1 in patient samples.** Having tested detection using the SARS-CoV-2 heat-inactivated virus and genomic RNA for crRNA optimization and LoD determination, we sought to validate the ENHANCEv1 in patient samples to determine whether it still maintains robust detection with clinical conditions. We tested a total of 62 nasal swab samples in which 31 samples were predetermined SARS-CoV-2 positive and the other 31 samples negative. Samples were randomly selected and blinded prior to viral RNA extraction. Similar to DETECTR[14], the RNase P gene was used as an internal control for all the nasal swab samples. For the real-time fluorescence-based reporter detection, the criterion for a positive sample is a change in fluorescence signal of 5-fold or greater compared to non-template control (NTC) samples within 20 min. Consistent with the

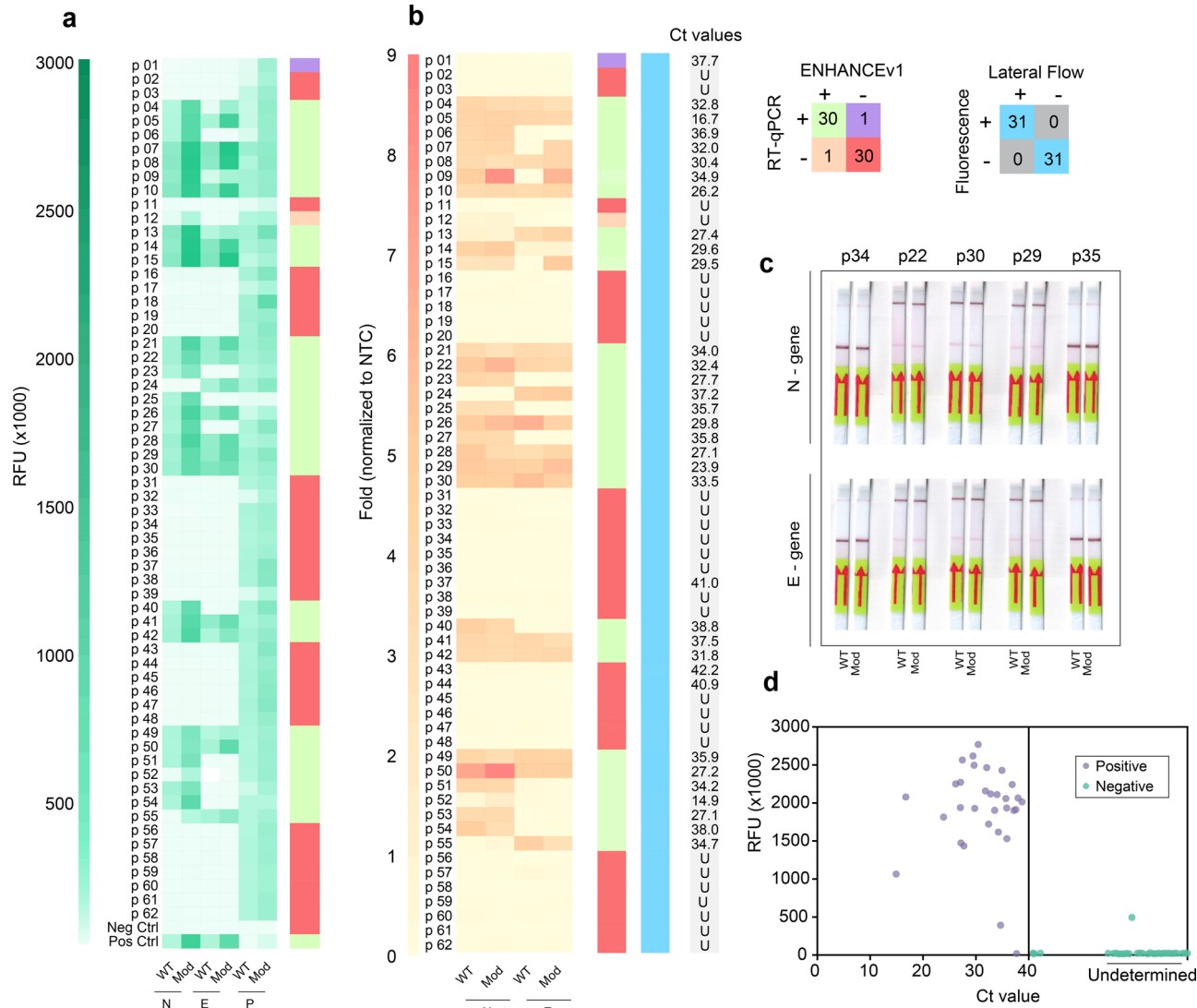

**Fig. 3 Clinical validation of ENHANCE. a** Fluorescence intensity of wild-type crRNA (crRNA-WT) and modified crRNA (crRNA-Mod) detecting SARS-CoV-2 RNA in 62 patient samples using the fluorescence-based reporter assay. Five microliters of extracted samples were used for the 30-min RT-LAMP step followed by the CRISPR detection assay. Fluorescence signals were taken at $t = 20$ min. **b** Lateral flow assay detecting SARS-CoV-2 RNA in the same 62 patient samples. A heat map shows the ratio of band intensity of the positive line (top band) to the control line (bottom band). Band intensities were analyzed by ImageJ. **c** Representation of lateral flow assay testing 62 patient samples. For full details, please see the supplementary figure S1. **d** Maximum fluorescence intensity of 62 patient samples against Ct values validated with RT-qPCR. For Ct values, the letter "U" denotes undetermined.

previous observation, the EHANCEv1 demonstrated a higher sensitivity across all positive samples at low magnesium concentrations in a shorter amount of time (Fig. 3a). Furthermore, the ENHANCEv1 achieved a 96.7% overall accuracy with a false positive rate and a false negative rate of 3.3% and 3.3%, respectively (Supplementary Fig. S2 and Supplementary Table S1). For lateral flow-based detection, the ENHANCEv1 showed 100% agreement with the real-time fluorescence-based reporter assay. Similarly, the false positive and false negative rates for this assay were both 3.3% (Fig. 3b–d and Supplementary Figs. S3, S4). Based on the results from ENHANCE, we observed that Ct values do not always correlate well with the fluorescence intensity from the CRISPR reaction (Fig. 3d). It is possibly due to an inhibitory effect on the CRISPR detection reaction by excessive amplified product. This observation corroborates a study by Li et al.[38]. in which they have noted a similar phenomenon. In addition, as the copy number of nucleic acid template decreases, there is an increasing variation in RT-LAMP reactions in terms of time to

reach signification amplicons, as demonstrated by Hardinge et al.[39].

**Lyophilized ENHANCEv2 reduces CRISPR reaction time**. We further developed a CRISPR-ENHANCEv2 by combining ENHANCEv1 with the mutated LbCas12a$^{D156R}$ and a dual reporter comprised of a fluorophore FAM, biotin, and a quencher. This reporter can be used as a fluorescence-based reporter as well as a lateral flow-based reporter, both in a single detection assay (Fig. 4a). We designed three different dual reporters and found that the dual reporter version 2 worked best in terms of sensitivity and produced low background on negative samples (Supplementary Table S2). A typical CRISPR reaction requires the complexation of crRNA and LbCas12a prior to the addition of a pre-amplified activator. Therefore, we initially sought to understand whether the complexation of crRNA and LbCas12a can be achieved and remain stable for a period of time

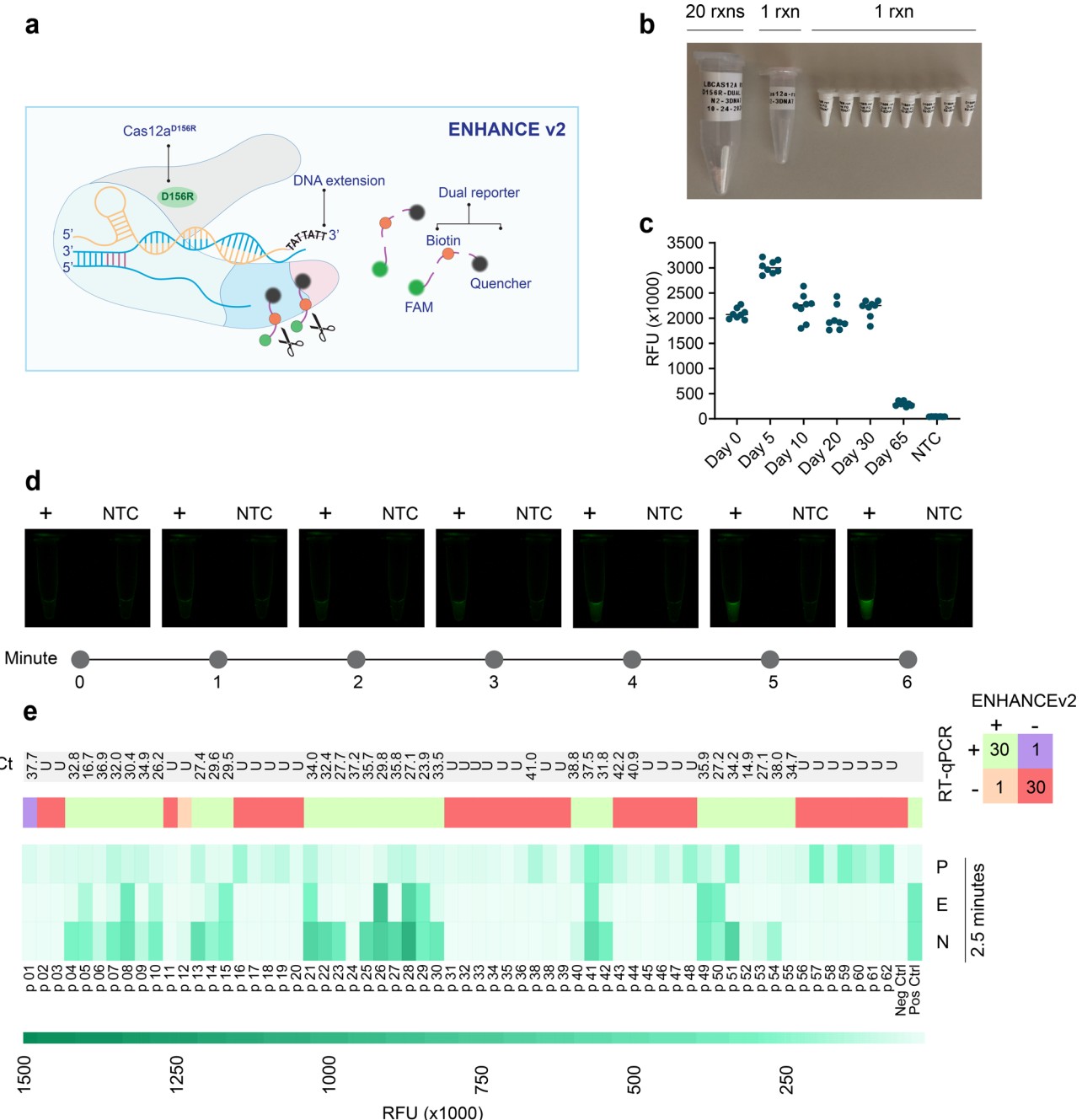

**Fig. 4 Clinical validation of ENHANCEv2. a** Schematic diagram of composition of ENHANCEv2 system. **b** Lyophilized samples in different pilot scales (20, 5, and 1 reactions) of CRISPR reaction used in ENHANCEv2. The CRISPR reaction contains pre-complexed crRNA:LbCas12a$^{D156R}$, dual reporter, and NEBuffer 2.1. Only RNase-free water and RT-LAMP products are added to activate the reaction. **c** Stability testing of lyophilized CRISPR reaction using ENHANCEv2 with respect to time. **d** Time-course observation of lyophilized CRISPR reaction. Positive samples can be seen by naked eyes under blue light within as low as 2 min. Samples were visualized in an Analytik Jena UVP Gelstudio system under blue light (wavelength range 460–470 nm) with SYBR green filter. **e** Fluorescence intensity of ENHANCEv2 detecting 62 patient samples at $t = 2.5$ min. For Ct values, the letter "U" denotes undetermined.

prior to testing. If successful, large batches of pre-complexed crRNA:LbCas12a can be manufactured to reduce reaction time and minimize the need for specialized training. We explored if lyophilization can accomplish this objective. Optimal concentrations of crRNA and LbCas12a$^{D156R}$ were pre-complexed in a mixture containing dual reporter and cleavage buffer. This combined mixture was prepared in different pilot scales: 1 reaction, 5 reactions, and 20 reactions (Fig. 4b). By conducting a time study, we observed that these lyophilized pre-complexes maintain high trans-cleavage activity compared to the liquid version and

remain stable for up to 30 days upon storage at room temperature (Fig. 4c and Supplementary Fig. S5). These results help enable scale-up manufacturing, ease of transportation, storage, and reduction in preparation time.

Surprisingly, we also observed a noticeable decrease in reaction time. Using lyophilized CRISPR reaction stored at −20 °C after 2.5 months for the fluorescence reporter assay, the fluorescence signal can be seen by naked eyes under blue light after 2 min (Fig. 4d). When applying the lyophilized ENHANCEv2 towards 62 patient samples, we were able to accurately

detect the same nasal swab samples with >5-fold change in fluorescence compared to NTC samples in 2.5 min (Fig. 4e and Supplementary Fig. S6), bringing a total reaction time to 33 min (30 min of RT-LAMP). This phenomenon strongly suggests that pre-complexed LbCas12a-crRNA promotes a faster and easier assay setup. Additionally, we have successfully lyophilized the RT-LAMP reactions, which in combination with the lyophilized CRISPR reaction could support ease in transportation and alleviate cold storage in remote areas (Supplementary Fig. S7).

## Discussion

In the wake of the COVID-19 outbreak, detection and vaccine developments become the two important pillars of research to eradicate the disease[40,41]. At the time of preparing this manuscript, the first vaccine developed by Pfizer has been granted emergency authorization by the FDA. This news gives hope to combat the COVID-19 disease that has already caused great damage to our community. However, it will take months for the vaccine to get in the hands of millions of people, especially in developing countries; therefore, COVID-19 testing remains a necessary tool not only for tracking the severity of the spread and following up the population post-vaccination but also for surveillance testing as a preventative measure against a future pandemic. Many CRISPR-based detection platforms such as DETECTR, SHERLOCK, and STOPCovid have emerged as an alternative to traditional RT-qPCR assays due to their rapid, affordable, and sensitive capabilities[14,18,34,35,42]. Here, we developed ENHANCEv2, an engineered CRISPR-based diagnostic platform to detect SARS-CoV-2 in patient samples with high sensitivity and specificity. Similar to the DETECTR system, ENHANCE utilized a robust trans-cleavage capability of Cas12a towards detection[13,14].

ENHANCEv1 consists of a chimeric 7-mer TA-rich extension the 3'-end of crRNA and wild-type LbCas12a. Our previous study has shown that this modified CRISPR-Cas system markedly enhances Lbcas12a trans-cleavage activity while slightly improving its specificity[24]. The theoretical limit of detection for ENHANCEv1 was determined to be 15 copies/μL and 25 copies/μL for N gene and E gene, respectively, achieving a high level of sensitivity in a sample volume as small as 5 μL with 95% accuracy. This engineered system also exhibited high specificity towards SARS-CoV-2 when tested against highly similar and commonly circulating pathogens. Finally, ENHANCEv1 was used to validate 62 patient samples in this work, with 60/62 patients in agreement with RT-qPCR results. One false negative and one false positive were observed in 31 negative and 31 positive samples for both fluorescence-based reporter and lateral flow assay, showing a total of 96.7% accuracy. We encountered some challenging results during clinical validation of ENHANCEv1 and ENHANCEv2 with 62 patient samples. Patient sample p01 was reported as negative upon arrival in our lab. The sample was also determined as negative with our ENHANCE assay, but revalidation with RT-qPCR assay indicated it as a positive sample. Therefore, we concluded that the sample was a false negative based on our analysis approach. For patient sample p43, both ENHANCE and RT-qPCR assays agreed that the sample was negative with SARS-CoV-2, but it was predetermined as a positive sample. Thus, we concluded this sample was negative. The patient sample p12 was predetermined to be a negative sample and was detected negative with the qPCR; however, it was consistently detected as positive by ENHANCEv1 and v2 methods. Therefore, we concluded it as a false positive. Finally, QC failure was observed while testing samples p32, p38, and p62 as fluorescence signals for the RNase P control gene did not pass the threshold requirement. However,

we were able to successfully detect these samples upon the second round of RNA extraction.

We developed ENHANCEv2 to further improve the testing capability of ENHANCEv1. ENHANCEv2 is comprised of a 7-mer DNA extension on the 3'-end of crRNA, a mutated LbCas12a[D156R], and a dual reporter that can be used for both fluorescence-based and lateral flow assays. The dual reporter enables detection of clinical samples in two different assay formats that can be potentially used for conducting a high-throughput lab-based and rapid point-of-care-based test using a single CRISPR kit. ENHANCEv2 was used to test the same 62 patient samples, showing 100% agreement with ENHANCEv1 while reducing CRISPR reaction time to as low as 3 min. Furthermore, unlike ENHANCEv1, the ENHANCEv2 does not require a crRNA/Cas pre-incubation step, saving 15–20 min of time and labor. These results were achieved by lyophilizing pre-complexed crRNA:Lb-Cas12a beforehand. The lyophilized version of ENHANCEv2 not only accelerates CRISPR reaction speed but also remains stable for several weeks upon storage at room temperature.

Similar to SHERLOCK and DETECTR, a major drawback to this technology is the possibility of carryover contamination. CRISPR-based detection system provides two checkpoints for detection: (1) an isothermal pre-amplification step such as RT-LAMP that converts SARS-CoV-2 genomic RNA into millions of dsDNA target copies and (2) a CRISPR reaction in which crRNA:LbCas12a complex targets the amplified dsDNA. Since this detection platform is not a continuous step, contamination might occur. Therefore, a proper reaction setup must be met. This major disadvantage motivated us to develop ENHANCEv2. We also incorporated UDG and dUTP in the CRISPR detection reactions. ENHANCEv2 uses RT-LAMP master mix containing UDG which has been shown to greatly eliminate carryover contamination[43]. We observed a higher LoD with the incorporation of UDG, but we saw much lower risks of false positives. Cleaning the area with RNase away, using filtered pipette tips, and preparing LAMP/RT-LAMP reactions within a nucleic acid workstation (with HEPA filtration and UV light irradiation) can further reduce contamination. For fluorescence detection, RNase P was used as a control for QC failure, and negative controls were used to ensure master mixes did not contain contamination. The lyophilized CRISPR version not only reduces the reaction time but also simplifies the needs for CRISPR reaction setup. The reaction can be reconstituted with the addition of RNase-free water and RT-LAMP products. The method further promotes large-scale manufacturing, eases the challenge of reagent storage and handling, and would enable easier deployment across the globe. Additional lyophilization of the RT-LAMP reaction may reduce contamination and allow for a more practical at-home testing solution.

The future of CRISPR-based diagnostic platforms is bright. In addition to SHERLOCK and DETECTR technologies receiving emergency use authorization by the FDA, recent collaboration of MilliporeSigma and Mammoth Biosciences brings forward scale-up of a new SARS-CoV-2 test, allowing for rapid turnaround time and massive testing capability[44]. Isothermal detection using CRISPR enables rapid results without the need for expensive equipment. However, since many CRISPR detection methods involve a highly sensitive pre-amplification, contamination remains a challenge. This problem is not specific to CRISPR tests as the gold standard RT-qPCR must also overcome this issue. We hope that the improved capability and stability of ENHANCEv2 will aid in global deployment towards curbing the COVID-19 pandemic.

## Data availability

All data associated with this study are in the main text and the Supplementary Materials. Supplementary Data 1 contains source data for the main figures in this manuscript.

Supplementary Data 2 contains oligonucleotide sequences used in the study. Additional data for the supplementary figures are available upon request.

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

## Acknowledgements

We are grateful for the members of Jain Lab for helpful advice and discussions. We thank the University of Florida (UF), the UF Health Cancer Center, and the Clinical and Translational Science Institute Repository (CTSI) for their support. We also thank Dr. David Ostrov and Dr. Cuong Nguyen for their support with obtaining the patient samples and for their discussions. Finally, we thank Dr. Whitney Stoppel and her lab in the Department of Chemical Engineering and Dr. Chrisopher Dervinis in the Forest Genomics Group at the UF for the use of the lyophilizers. The following reagents were obtained through BEI Resources, NIAID, NIH: Genomic RNA from the Middle East Respiratory Syndrome Coronavirus (MERS-CoV), EMC/2012, NR-45843; Quantitative PCR (qPCR) Control RNA from Inactivated SARS Coronavirus, Urbani, NR-52346; Genomic RNA from Human Coronavirus (HCoV), NL63, NR-44105; Genomic RNA from SARS-Related Coronavirus 2, Isolate Hong Kong/VM20001061/2020, NR-52388. The following reagent was deposited by the Centers for Disease Control and Prevention and obtained through BEI Resources, NIAID, NIH: Genomic RNA from SARS-Related Coronavirus 2, Isolate USA-WA1/ 2020, NR-52285; Quantitative PCR (qPCR) Control RNA from Heat-Inactivated SARS-Related Coronavirus 2, Isolate USA-WA1/2020, NR-52347; SARS-Related Coronavirus 2, Isolate USA-WA1/2020, Heat Inactivated, NR-52286. The following reagent was obtained from Dr. Maria R. Capobianchi through BEI Resources, NIAID, NIH: Genomic RNA from SARS-Related Coronavirus 2, Isolate Italy-INMI1, NR-52498. Funding: This work was financially supported by funds from the UF, UF Herbert Wertheim College of Engineering, Florida Breast Cancer Foundation (AGR00018466), United States-India Science & Technology Endowment Fund: COVID-19 Ignition Grants (USISTEF/COVID-I/247/2020), National Institute of Health (NIAID R21AI156321), and Centers for Disease Control and Prevention (U01GH002338).

## Author contributions

P.K.J., L.T.N., and S.R.R. designed the experiments. L.T.N., S.R.R., and B.L.M.P. performed the experiments. L.T.N., S.R.R., B.L.M.P., and B.T.S. performed data analysis.

L.T.N., S.R.R., B.L.M.P., and B.T.S. wrote the initial manuscript. The manuscript was edited by P.K.J. and revised by all members.

## Competing interests

P.K.J. and L.T.N. are listed as inventors on the multiple patent applications related to the content of this work. P.K.J. is a co-founder of Genable Biosciences, LLC. The remaining authors declare no competing interests.
