## [Peer Review File · Communications Medicine]

Reviewers' comments:

Reviewer #1 (Remarks to the Author):

This paper by Nguyen et al. describes the clinical testing of a previously described Cas12-based diagnostic method called ENHANCE. The method performs well in terms of clinical sensitivity, which is great to see. The analytical sensitivity data for N2 are messy and need improvement or just need to be repeated; this seems to be specific to the N2 amplicon as the other ones (while they are less sensitive) don't appear to have this issue. In addition, I am a bit concerned that ENHANCE v2 does not appear to offer much in the way of advantages over v1.

Major comments:

1. In Fig 2b, some of the negative controls (0 copies of input) are positive. This is concerning, for example both the 3.2 copies / ul and 0 copies / ul appear to have one of three replicates that are positive. I am not convinced that the assay is performing consistently here.

2. The performance of N2 is worse at 0.4 copies per ul than at 0.2 copies per ul (Fig. 2e). Moreover, the calculation in Table S1 would suggest that the method is capable of detecting a single molecule (5 ul of input into the LAMP reaction at 0.2 copies per ul) in 19/20 replicates. This violates the Poisson distribution - one would expect to detect a single molecule only 37% of the time, regardless of the method. This would suggest that either there is an error with the concentration measurements, or that there is some contamination causing detection at this lower concentration.

Another issue is that the method works less well with twice as much input; only 16/20 replicates were detected from the higher (0.4 copy per microliter) condition.

Given these data, it is not accurate to claim a sensitivity of 0.2 copies per ul. Furthermore, it would be very helpful to see some negative controls from this experiment. In light of these discrepancies, the authors should rerun their limit of detection experiment for N2.

3. The performance of the dual reporter for lateral flow detection appears to be poor; there are clear bands in the negative controls in Fig. S5b. This undermines the novelty of ENHANCE v2, as it consists of using a mutant Cas12, and a dual reporter. More broadly I am left wondering: what are the advantages of v2 relative to v1? The performance appears to be identical, the only advantage I can see is the lyophilization (but is this really specific to v2? is the mutant Cas12 more stable after lyophilization?). I would like to see the authors demonstrate that their dual reporter (or mutant Cas12) significantly improves the v2 method, relative to v1; otherwise it is not clear why one should use v2 instead of v1.

Minor comments:

1. The authors do not use an extraction/process control for their clinical sample testing; this could explain why they were unable to detect one of the patient samples.

2. There are a few typos/grammatical errors in the manuscript. Please fix these.

3. The authors demonstrate lyophilization of the CRISPR reagents for ENHANCE - but not the LAMP reagents. Can the LAMP reagents also be lyophilized? If not, the assay would still require a cold chain.

Reviewer #2 (Remarks to the Author):

Nguyen et al is a follow-up study to the authors' recent publication on the improvement of Cas12-based detection through the use of crRNA with DNA extension, called ENHANCE. The present work contains clinical data for ENHANCE, which is now extended to the detection of the SARS-CoV-2 E gene and human RNase P gene, in addition to the previously reported detection of the N gene. In the present work, the authors also develop ENHANCEv2, with the following modifications to ENHANCE: 1) the use of a Cas12a mutant; 2) the preparation of the CRISPR reaction in lyophilized form; and 3) the use of a dual-functioned reporter for fluorescence and lateral-flow detection.

Overall this is a well-written study, but the improvements presented are unfortunately marginal. In fact, it was not clear to me that there is any tangible improvement of ENHANCEv2 vs the original ENHANCE. The Cas12a-D156R mutant does not improve the detection sensitivity compared to wild-type LbCas12a in any of the assays shown by the authors. Freeze-drying Cas-crRNA complexes, while definitely helpful for transportation, has been demonstrated with other Cas-crRNA complexes, and can likely be performed with wild-type LbCas12a of ENHANCE. The dual-functioned reporter is useful but not game-changing.

Despite the lack of much technological improvement, I could see the value of the study as a validation work with strong clinical implications. The author's previous work on crRNA engineering is exciting, and it is satisfying to see the technology's clinical performance here. To strengthen the validation work, the authors should report threshold cycle values of each clinical sample (by RT-PCR) used for clinical validation so that readers can critically judge the clinical sensitivity of the technology. Moreover, it would be nice to see a direct comparison of ENHANCE vs ENHANCEv2 in key performance parameters e.g. limit of detection, reaction time (time to positive results). The naked-eye positive results under 2 min of Fig 4d is promising, but not compelling without direct comparison to ENHANCE and more information on viral loads of the sample.

Minor comments:

- Can the authors speculate further on patient sample p12, which gives negative results on RT-PCR but positive on ENHANCE(v2)? What would be the possible explanations?
- Page 6: can the authors provide references to support the claim "correlation between cis-cleavage and trans-cleavage capabilities" of Cas enzymes?
- Page 6: the authors mentioned LoD was determined for 6 crRNAs, but only showed results from 4 crRNAs
- Page 7: ENHANCEv1 is misspelled
- Page 8: the authors referred to Figure 3F but I do not see this subfigure in Figure 3. Labels A and B were also missing from Figure 3.

Reviewer #3 (Remarks to the Author):

The work of Ngyun et al provides clinical validation of the CRISPR/Cas12a system for the detection of SARS-CoV2. This work is well-written and timely.
I have the following comments on this manuscript.

1- Although the authors have shown that the modified crRNA exhibits much higher trans cleavage activity, it is not clear what impact this enhancement has on the sensitivity or specificity of the assay. In other words, would this enhancement lead to improvements in the sensitivity/ specificity, or just it may lead to a reduction of the assay time? This point needs to be sufficiently addressed.

2- Since this work aims primarily on the validation of the CRISPR/Cas12 assay. I would expect that the improved system tested on an expanded number of clinical samples with different ct values. I recommend the authors use clinical samples of ct values below and above 30 to explore the concordance of the sensitivity of this assay with the RT-qPCR. Such quantitative data are needed to substantiate the message of the manuscript and reflect that these improvements influence the performance of the assay on the clinical samples. Not just the strength of the signals.

3- I wonder whether the authors have tested their system in a one-pot assay which is of great importance for the development of the POC system. The authors should sufficiently discuss what could be done to limit the cross-contamination.

4- I think the authors need to modify the title to reflect the message of the manuscript which is the clinical validation of the system. More quantitative data are required with comparisons with RT-qPCR with different ct values.

5- Authors need to cite other works on the CRISPR/Cas12 system including “iSCAN: An RT-LAMP-coupled CRISPR-Cas12 module for rapid, sensitive detection of SARS-CoV-2” by Ali et al 2020.

6- Authors need to provide one paragraph as an update on the status of CRISPR technologies for SARS-CoV2 detection and current challenges and context and the importance of this work.

Minor

1- Please fix “ we developed a lyophilized a version of ENHANCEv2 system” to “ we developed a lyophilized version of ENHANCEv2 system”

2-

3- Please fix “ The ENHANCE system works robustly with a wide range of a magnesium concentration (3mM- 13mM)” to “ The ENHANCE system works robustly with a wide range of magnesium concentrations (3mM- 13mM)”

Reviewer #4 (Remarks to the Author):

It was an interesting work that the authors introduce ENHANCE system (V1 and V2), both showed a significant increase in sensitivity while maintaining specificity even at low magnesium concentrations. ENHANCEv2 system, which utilizes modified crRNAs from ENHANCE, a mutated LbCas12a protein (Cas12aD156R) 22-25 to further amplify the signal, and a dual reporter construct that allows each sample to be read using a fluorescence-based and a lateral flow assay format in the same reaction. The lyophilized CRISPR reaction in ENHANCEv2 occurs at an accelerated rate and preserves for a long period of time upon room temperature storage. The manuscript is well organized and easy to follow, but I still have some comments as follows:

1st, As the authors state the limit of detection (LoD) the ENHANCE platform (crN2) is 0.2 copies/ μL , while, the LoD of the US FDA EUA-approved CDC assay for detection of SARS-CoV-2 is 1 copies/ μL . However, ENHANCE platform add 5 μL template to reaction, and CDC assay add 2 μL template to reaction. Actually, both methods have same sensitivity, about 1~2 copy/rxn. LoD of the ENHANCE platform described in this manuscript will confused the readers that ENHANCE has higher sensitivity than CDC assay.

2nd, in Discussion section. Authors discussed three samples of conflicting results. Do you know what is the pre-determined method? QPCR or clinical diagnosis? QPCR should not be the only diagnostic criteria. Authors should be make sure the sample type, not only based on QPCR.

Patient sample p01 was pre-determined as negative upon arrival in your lab, it is also indicated as negative with your ENHANCE assay, but revalidation with RT-qPCR assay indicated it as a positive sample. Since ENHANCE has same sensitivity with QPCR, why you concluded that this sample was a false negative based on your analysis approach, whether it is possible that QPCR is wrong?

The patient sample p12 was predetermined to be a negative sample and was detected negative with the qPCR, however, it was consistently detected as positive by ENHANCE v1 and v2 methods. Is it possible that ENHANCE is right and p12 is false negative?

3rd, In Main section, the third paragraph. Authors mentioned that Cas12a-based DETECTR has a limit of detection of around 20 copies/ μL , while SHERLOCK is based on Cas13a and is timeconsuming. It is wrong. I read the paper "CRISPR-Cas12-based detection of SARS-CoV-2", the LoD of DETECTR is 10 copies/ μL , and 2 μL template add to the reaction, 20 copy/rxn. Please make sure the LoD of DETECTR and SHERLOCK.

4th, In Main section, the third paragraph, authors stated that SHERLOCK utilizing Recombinase Polymerase Amplification (RPA) and has a limit of detection of around 6.75 copies/ μL , while DETECTR utilizing Reverse Transcription Loop Mediated Isothermal Amplification (RT-LAMP), has a limit of detection of around 20 copies/ μL . But the last sentence of this paragraph, author write "The RT-LAMP, performed at a constant temperature of 60-65 $^{\circ}\text{C}$, is significantly less timeconsuming due to its "cauliflower-like" logistic growth amplification pattern. It is also more sensitive than RPA and can be easily adopted without supply-chain issues". It is could be confused.

5th, ENHANCEv2 uses RT-LAMP master mix containing UDG which has been shown to greatly eliminate carryover contamination, this paper should be cited. Analytical Chemistry 2019 91 (17), 11362-11366

General response to the reviewers:

We appreciate the reviewers' feedback on our manuscript titled "Engineered CRISPR/Cas12a enables rapid SARS-CoV-2 detection".

- We have addressed all the comments below with responses marked in blue.
- Changes to the manuscript are highlighted in yellow.

Reviewer #1 (R1) (Remarks to the Author):

This paper by Nguyen et al. describes the clinical testing of a previously described Cas12-based diagnostic method called ENHANCE. The method performs well in terms of clinical sensitivity, which is great to see. The analytical sensitivity data for N2 are messy and need improvement or just need to be repeated; this seems to be specific to the N2 amplicon as the other ones (while they are less sensitive) don't appear to have this issue. In addition, I am a bit concerned that ENHANCE v2 does not appear to offer much in the way of advantages over v1.

Major comments:

R1.1. In Fig 2b, some of the negative controls (0 copies of input) are positive. This is concerning, for example both the 3.2 copies / ul and 0 copies / ul appear to have one of three replicates that are positive. I am not convinced that the assay is performing consistently here.

Response to R1.1. Thank you for the great feedback. We agree with the reviewer that false positives in the fig. 2b might have been due to carryover contamination, since N2 LAMP primers exhibit high sensitivity. At the time of testing, we used the RT-LAMP master mix that did not contain Uracil DNA glycosylase (UDG), which led to significant contamination issue. To eliminate contamination, we now incorporated UDG in our protocol and repeated the experiment to determine the LoD of the N2 gene. Additionally, all the experiments we performed to investigate the reviewers' comments were carried out with UDG. Notably, we observed consistent performance of N2 LAMP primers after the changes. Changes to fig. 2 are shown in the revised main text and the previous data is now in the SI section, fig. S2. In addition, we have commented in the caption of the figure as follows:

"Fig. 2. Limit of Detection and Exclusivity Testing of ENHANCE. (A), (B), (C), and (D) Fluorescence intensities in RFU of wild-type crRNA (crRNA-WT) and modified crRNA (crRNA-Mod) in serially diluted samples containing quantitative genomic RNA of SARS-CoV-2 (n = 3). The RT-LAMP amplification was performed with the addition of UDG since its high sensitivity easily leads to carryover contamination. (E) and (F) 2/3 of samples detected in (A), (B), (C), and (D) were subject to be tested with 20 different serial dilutions around the estimated LoD for N gene and E gene. (G) and (H) Scanned images of samples in (E) and (F), respectively. (I) Exclusivity testing of ENHANCE against highly similar and commonly circulating pathogens. Genomic RNAs were diluted down to a concentration of around 1500 copies/ μ L in TE buffer followed by 30-minute

RT-LAMP. 2 μ L of the RT-LAMP product was used for the CRISPR detection assay. Fluorescence signals were taken at $t = 20$ minutes. All samples with Ct value ranges were obtained from Zepetometrix.”

R1.2. The performance of N2 is worse at 0.4 copies per ul than at 0.2 copies per ul (Fig. 2e). Moreover, the calculation in Table S1 would suggest that the method is capable of detecting a single molecule (5 ul of input into the LAMP reaction at 0.2 copies per ul) in 19/20 replicates. This violates the Poisson distribution - one would expect to detect a single molecule only 37% of the time, regardless of the method. This would suggest that either there is an error with the concentration measurements, or that there is some contamination causing detection at this lower concentration.

Another issue is that the method works less well with twice as much input; only 16/20 replicates were detected from the higher (0.4 copy per microliter) condition.

Given these data, it is not accurate to claim a sensitivity of 0.2 copies per ul. Furthermore, it would be very helpful to see some negative controls from this experiment. In light of these discrepancies, the authors should rerun their limit of detection experiment for N2.

Response to R1.2. We thank the reviewer for the great feedback. We agree with the reviewer regarding the 0.2 copies per μ L limit of detection for N2 gene. Despite reporting true results of our experiments, we speculated that contamination might have caused inconsistency in our experiments. We initially divided our validation study in multiple stages. The LoD determination experiments were carried out first without the incorporation of UDG in the experiments.

With that being said, we have re-performed the LoD experiments for the two most sensitive genes used for detecting SARS-CoV-2 in patient samples, N2 and E2 with the incorporation of UDG this time. The data have been consistent, and we observed an increase in the LoD of N2 and E2 gene with the LoD of 15 copies/ μ L and 25 copies/ μ L, respectively. The Fig. 2 in the main text has been updated with appropriate panels. We also observed that incorporation of UDG greatly eliminated cross-over contamination, but at the same time resulted in less sensitivity of the RT-LAMP reaction. This phenomenon is possibly due to the addition of dUTP (substrate of cleavage of UDG).

Changes to the main text and the supplementary figures are as follows:

“Fig. 2. Limit of Detection and Exclusivity Testing of ENHANCE. (A), (B), (C), and (D) Fluorescence intensities in RFU of wild-type crRNA (crRNA-WT) and modified crRNA (crRNA-Mod) in serially diluted samples containing quantitative genomic RNA of SARS-CoV-2 (n = 3). The RT-LAMP amplification was performed with the addition of UDG since its high sensitivity easily leads to carryover contamination. (E) and (F) 2/3 of samples detected in (A), (B), (C), and (D) were subject to be tested with 20 different serial dilutions around the estimated LoD for N gene and E gene. (G) and (H) Scanned images of samples in (E) and (F), respectively. (I) Exclusivity testing of ENHANCE against highly similar and commonly circulating pathogens. Genomic RNAs were diluted down to a concentration of around 1500 copies/μL in TE buffer followed by 30-minute RT-LAMP. 2 μL of the RT-LAMP product was used for the CRISPR detection assay. Fluorescence signals were taken at t = 20 minutes. All samples with Ct value ranges were obtained from Zepetmetrix.”

In addition, changes to the LoD have been updated in the main text (highlighted in yellow). The LoD determination without UDG was moved to the supplementary information as follows:

Fig. S2. LoD determination of N2 gene when UDG is not incorporated in the pre-amplification step. (A) Estimated LoD experiments were performed as described in figure 2 in the main text. An estimated LoD was determined based on the lowest copies/μL with 2/3 replicates detected positive. (B) 20 replicates with the copies/μL at 1X and 2X times the estimated LoD in (A) were repeated to confirm the final LoD. (C) Fluorescence image of samples in (B).

We have also commented on this matter in the main text as follows:

We noticed a significant difference in LoD between the pre-amplification reaction with the incorporation of Uracil DNA glycosylase (UDG) and one without UDG. The above reported LoD for N2 and E2 genes were determined using UDG, which seemed to be higher than the LoD for the same two genes in the absence of UDG. We detected 19/20 replicates to be positive at 0.2 copies/μL and 7.9 copies/μL, respectively (fig. S2). The LoD for N2 violated the Poisson distribution possibly due to carryover contamination.

R1.3. The performance of the dual reporter for lateral flow detection appears to be poor; there are clear bands in the negative controls in fig. S5b. This undermines the novelty of ENHANCE v2, as it consists of using a mutant Cas12, and a dual reporter. More broadly I am left wondering: what are the advantages of v2 relative to v1? The

performance appears to be identical, the only advantage I can see is the lyophilization (but is this specific to v2? is the mutant Cas12 more stable after lyophilization?). I would like to see the authors demonstrate that their dual reporter (or mutant Cas12) significantly improves the v2 method, relative to v1; otherwise, it is not clear why one should use v2 instead of v1.

Response to R1.3. We are grateful for the thoughtful comments. We believe that this is a valid concern, and that the dual reporter has high background on the negative samples. To address this issue, we have spent a lot of time and effort designing two additional dual reporters and testing them out. The sequence information of these two dual reporters have been added to the supplementary information. We observed that newly designed dual reporter version 2 displayed the lowest background on negative samples and achieved comparable performance as the single reporter. We have tested this dual reporter v2 in patient samples. Changes to fig. S5b has been updated. We have also validated this dual reporter version 2 on fluorescence-based assay and observed great performance. Our dual reporters may seem simple but are elegant design with unique multi-modal detection capabilities that has never been reported before. They are advantageous for both the manufacturers and the users. The manufactures can lyophilize a single kit with dual reporters instead of two separate kits for different settings (lab or point-of-care/resource-limited). The users can choose a mode depending on the availability and can expect the same results irrespective of the mode of use. Since fluorescence is non-destructive, one can visualize/measure the fluorescence and then spot the same sample after a simple dilution on a lateral flow strip without having to run the entire protocol again.

Please note that our previous work on ENHANCEv1 (Nguyen et al., *Nat. Comm.*, 2020) was preliminary development of the detection technology for N gene using genomic RNA of SARS-CoV-2 and did not include any live or inactivated viruses, respiratory pathogens, clinical isolates, clinical samples, other SARS-CoV-2 genes, or internal control genes. Here, we optimized, characterized, and clinical validated ENHANCEv1 for the first time, for which the major steps (not emphasized in the manuscript) included obtaining an IRB approval and getting access to the inactivated viral particles and clinical samples in the midst of the pandemic. We first optimized and validated multiple RT-LAMP primers and crRNAs spanning across the SARS-CoV-2 genome (RdRp, N, E) and internal control (human RNase P) genes with ENHANCEv1. Then we optimized various viral extraction methods from clinical matrices, obtained and tested including 3 clinical isolates for inclusivity, 31 respiratory pathogens for exclusivity, various inactivated viral particles for LOD, and validated 62 clinical samples with RT-PCR to determine the Ct values, and then validated them with both fluorescence and lateral flow ENHANCEv1 platforms highlighted in the manuscript. We have now included a comparison of ENHANCEv1 and ENHANCEv2 with DETECTR in the fig. S7.

In terms of the advantages of v2 over v1, we agree with the reviewer that the ENHANCEv2 is slightly improved over ENHANCEv1 but not significantly. However, our

rationale for this development is to help put forward the CRISPR diagnostics field into practice as many other CRISPR pioneers are doing. Additionally, validation of patient samples in this study further solidified this technology. Therefore, our primary goals were to move towards developing rapid and sensitive CRISPR detection reagents for home-based testing. As you can see with the formulation in ENHANCEv2, the time to detection was significantly shortened to 2.5 minutes compared to 15 minutes in ENHANCEv1 (fig. 3 vs. fig. 4). Additionally, as pointed out by the reviewer, ENHANCEv2 was lyophilized without any pre-complexing step of Cas12a:crRNA, which shortens the time to detection even further. The idea of lyophilization did not come to us when we developed ENHANCEv1, because at that time we were interested in understanding how Cas12a reacts to extended crRNAs. Therefore, we were excited to apply lyophilization to ENHANCEv2 so that it can simplify the hand-on time and skills required to perform the experiment.

	DETECTR	ENHANCEv1	ENHANCEv2
	Chen et al., Science, 2018	Nguyen et al., Nat. Comm., 2020 Nguyen et al., Methods, 2021	This manuscript Nguyen et al., MedRxiv, 2020
crRNA	Wild-type (WT)	Engineered: 7-nt DNA on 3' end of crRNA (3'-DNA7)	Engineered: 3'DNA7
Cas	LbCas12a (WT)	WT	LbCas12a-D156R mutant
Kcat/Km (dsDNA)	$1.7 \times 10^7 \text{ s}^{-1} \text{ M}^{-1}$	$5.1 \times 10^7 \text{ s}^{-1} \text{ M}^{-1}$ (3.2-fold higher)	---
Pre-amplification	DNA: RPA	DNA: RPA RNA: RT-RPA & RT-LAMP	RNA: RT-LAMP
Targets	HPV16 & HPV18	HIV, HCV, PCA3, & SARS-CoV-2	SARS-CoV-2
Clinical validation	HPV16 (100% accuracy) HPV18 (92% accuracy)	None	SARS-CoV-2 (97% accuracy, 97% sensitivity, & 97% specificity)
LOD range	pM (-pre-amplification) aM (+pre-amplification)	fM (-pre-amplification) aM (+pre-amplification)	0.2-7.9 copies/ μL (-UDG) 15-25 copies/ μL (+UDG)
Assay time	60-120 min, 37°C (RPA+CRISPR)	30 min, 65°C (RT-LAMP) + 10-20 min, 37°C (CRISPR)	30 min, 65°C (RT-LAMP) + 3 min, 37°C (CRISPR)
Readout mode	Single mode: Fluorescence (FL)	Single mode: FL or lateral flow assay (LFA)	Dual mode reporters developed: FL+LFA in the same reaction
Key equipment	FL mode: Heater & plate reader	FL mode: Heater & plate reader LFA mode: Heater	FL mode: Heater & 460 nm LED LFA mode: Heater
Sample type	DNA & anal swabs	DNA, RNA, or heteroduplex (first-reported), & synthetic urine	RNA & nasal swabs
Extraction	Crude DNA extraction	Purified DNA and RNA	DNA extraction buffer at 15 min @ 65°C + 2 min @ 98°C & Maxwell for clinical samples.
Cold chain	Freezer required	Freezer required	ENHANCEv2 components lyophilized. Stable for 30 days at 25°C. RT-LAMP not lyophilized.
Reagent cost per reaction	<\$1 (F)	<\$1 (F), <\$3 (LFA)	<\$1 (F), <\$3 (LFA)
	First study on Cas12a detection. Several applications including COVID-19 diagnostics reported.	First study on enhancing collateral activity of a Cas by engineering crRNAs & enhancing sensitivity.	First study on developing and validating dual mode reporters. Lyophilized, stable for 30 days.

Fig. S7. Comparison of Cas12a-based DETECTR, ENHANCEv1, and ENHANCEv2.

The lyophilization is not specific to ENHANCEv2 as any stable protein in solution containing low concentration of glycerol can be freeze-dried. Initially, we were not sure if the mutant Cas12a-D156R was able to maintain its activity after lyophilization since it is a fairly unstable protein, and the process of free-drying usually takes more than 3 days to complete. Surprisingly, this mutant Cas12a is stable, and its performance is comparable to its form in solution (fig. S5). We thank the reviewer for the great insight. We have been pushing ourselves hard in the past three months since initial submission to improve the dual reporter, re-perform all the crucial experiments, and took our time to value what is important in our study. We envision that the CRISPR diagnostics is going to be put in practice with great capacity like traditional qPCR methods not in a distant future. We are hopeful that our little contribution, along with many great contributions from the scientific community, we will be able to see massive CRISPR surveillance testing in the next year or two.

Minor comments:

R1.4. The authors do not use an extraction/process control for their clinical sample testing; this could explain why they were unable to detect one of the patient samples.

Response to R1.4. This is a great suggestion, and we believe that it will improve our technique in working with detection-related projects. Like others, we used RNase P to keep track of the extraction quality control process; however, this might have posed issues as low-quality samples can only be determined after the detection for RNase P, which is time-consuming. Another way to overcome this issue is to add an exogenous gene fragment such as GFP in the sample prior to extraction to monitor the quality control. We will keep this suggestion in mind for our future projects. Thank you!

R1.5. There are a few typos/grammatical errors in the manuscript. Please fix these.

Response to R1.5. Thank you for pointing this out. We have thoroughly examined the typos/grammatical errors among the co-authors and have fixed them.

R1.6. The authors demonstrate lyophilization of the CRISPR reagents for ENHANCE - but not the LAMP reagents. Can the LAMP reagents also be lyophilized? If not, the assay would still require a cold chain.

Response to R1.6. Yes, LAMP reagents can also be lyophilized. Please refer to recent publications describing the lyophilization process of LAMP reagents such as *Paik, I et al.* titled “multi-modal engineering of Bst DNA polymerase for thermostability in ultra-fast LAMP reactions” from Ellington Lab.

We have produced bst polymerase in-house, and we are glad to tell the reviewer that it works well. We are currently formulating the storage buffer suitable for freeze-drying process. However, the experiments in this study were carried out using commercially available LAMP reagents from New England Biolabs before we figured all this out; we did not include here, but we are hopeful to have this lyophilization process detailed in our next projects.

Reviewer #2 (R2) (Remarks to the Author):

R2.1. Nguyen et al is a follow-up study to the authors' recent publication on the improvement of Cas12-based detection through the use of crRNA with DNA extension, called ENHANCE. The present work contains clinical data for ENHANCE, which is now extended to the detection of the SARS-CoV-2 E gene and human RNase P gene, in addition to the previously reported detection of the N gene. In the present work, the authors also develop ENHANCEv2, with the following modifications to ENHANCE: 1) the use of a Cas12a mutant; 2) the preparation of the CRISPR reaction in lyophilized form; and 3) the use of a dual-functioned reporter for fluorescence and lateral-flow detection.

Overall this is a well-written study, but the improvements presented are unfortunately marginal. In fact, it was not clear to me that there is any tangible improvement of ENHANCEv2 vs the original ENHANCE. The Cas12a-D156R mutant does not improve the detection sensitivity compared to wild-type LbCas12a in any of the assays shown by the authors. Freeze-drying Cas-crRNA complexes, while definitely helpful for transportation, has been demonstrated with other Cas-crRNA complexes, and can likely be performed with wild-type LbCas12a of ENHANCE. The dual-functioned reporter is useful but not game-changing.

Despite the lack of much technological improvement, I could see the value of the study as a validation work with strong clinical implications. The author's previous work on crRNA engineering is exciting, and it is satisfying to see the technology's clinical performance here. To strengthen the validation work, the authors should report threshold cycle values of each clinical sample (by RT-PCR) used for clinical validation so that readers can critically judge the clinical sensitivity of the technology. Moreover, it would be nice to see a direct comparison of ENHANCE vs ENHANCEv2 in key performance parameters e.g., limit of detection, reaction time (time to positive results). The naked-eye positive results under 2 min of Fig 4d is promising, but not compelling without direct comparison to ENHANCE and more information on viral loads of the sample.

Response to R2.1. We thank the reviewer for the valuable feedback. We have now included a comparison of ENHANCEv1 and ENHANCEv2 with DETECTR in the fig. S7. We agree with the reviewer that the Cas12a-D156R variant does not significantly improve the reaction rate compared the wild-type Cas12a. However, we would like to point out rationality of the version of ENHANCEv2:

1. LbCas12a has been widely used for nucleic acid detection in addition to AsCas12a as these two effector proteins show robust trans-cleavage activity. It is quite an effort to improve the reaction rate for a Cas12a protein that is already fast. ENHANCEv1 accomplished that by the addition of TA-rich extension on the

3'-end of the crRNA, which is simple yet quite effective and does not require labor-intensive investigation into the protein engineering. We came up with ENHANCEv2 to further enhance ENHANCEv1 with the expectation that the fold change in fluorescence signal would not be significant.

2. With that being said, we developed ENHANCEv2 to move forward with rapid and portable detection of nucleic acids. As you can see from fig. 3 and 4 in the main text, the time to detection of ENHANCEv2 is shortened to around only 3 minutes compared to around 15 minutes of ENHANCEv1. This is significant in moving the CRISPR diagnostics towards home-based testing in the future.
3. Although you see ENHANCEv2 is marginally better, validation of patient samples for both ENHANCEv1 and ENHANCEv2 was time-consuming but was accomplished. In our previous publication of ENHANCE, no patient samples have been tested. Per patient sample, there is only so much that can be extracted to validate each SARS-CoV-2 gene with both lateral flow and fluorescence-based detection assays.
4. Regarding the freeze-drying of Cas-crRNA complex, we agree with the reviewer that this process has been demonstrated for other Cas-crRNA complexes and can be used for the wild-type LbCas12a. However, this aspect of novelty is not the point of our study. We sought to show this freeze-drying process is practical, so that we can move the CRISPR-based diagnostic technology towards home-based testing. Additionally, freeze-drying process can differ depending on the formulation. We have spent quite some time to formulate the crRNA-Cas12a mixture to make sure that they are well-complexed and stable enough for a long duration. In addition, we also investigated the freeze-drying process in which the mutant LbCas12a-D156R was observed to be stable.
5. Thank you to the reviewer for the great suggestion on including the Ct value of each sample used for clinical validation. We have updated with the Ct value in Figs. 3a, 3b, and 4e in the main text.
6. We have also created a comparison chart of ENHANCEv1 and ENHANCEv2 vs. DETECTR for reference in the supplementary information (Table S7).
7. Please note that our previous work on ENHANCEv1 (Nguyen et al., *Nat. Comm.*, 2020) was preliminary development of the detection technology for N gene using genomic RNA of SARS-CoV-2 and did not include any live or inactivated viruses, respiratory pathogens, clinical isolates, clinical samples, other SARS-CoV-2 genes, or internal control genes. Here, we optimized, characterized, and clinical validated ENHANCEv1 for the first time, for which the major steps (not emphasized in the manuscript) included obtaining an IRB approval and getting access to the inactivated viral particles and clinical samples in the midst of the pandemic. We first optimized and validated multiple RT-LAMP primers and crRNAs spanning across the SARS-CoV-2 genome (RdRp, N, E) and internal control (human RNase P) genes with ENHANCEv1. Then we optimized various viral extraction methods from clinical matrices, obtained and tested including 3 clinical isolates for inclusivity, 31 respiratory pathogens for exclusivity, various

inactivated viral particles for LOD, and validated 62 clinical samples with RT-PCR to determine the Ct values, and then validated them with both fluorescence and lateral flow ENHANCEv1 platforms highlighted in the manuscript. We have now included a comparison of ENHANCEv1 and ENHANCEv2 with DETECTR in the fig. S7.

	DETECTR	ENHANCEv1	ENHANCEv2
	Chen et al., Science, 2018	Nguyen et al., Nat. Comm., 2020 Nguyen et al., Methods, 2021	This manuscript Nguyen et al., MedRxiv, 2020
crRNA	Wild-type (WT)	Engineered: 7-nt DNA on 3' end of crRNA (3'-DNA7)	Engineered: 3'DNA7
Cas	LbCas12a (WT)	WT	LbCas12a-D156R mutant
Kcat/Km (dsDNA)	$1.7 \times 10^7 \text{ s}^{-1} \text{ M}^{-1}$	$5.1 \times 10^7 \text{ s}^{-1} \text{ M}^{-1}$ (3.2-fold higher)	---
Pre-amplification	DNA: RPA	DNA: RPA RNA: RT-RPA & RT-LAMP	RNA: RT-LAMP
Targets	HPV16 & HPV18	HIV, HCV, PCA3, & SARS-CoV-2	SARS-CoV-2
Clinical validation	HPV16 (100% accuracy) HPV18 (92% accuracy)	None	SARS-CoV-2 (97% accuracy, 97% sensitivity, & 97% specificity)
LOD range	pM (-pre-amplification) aM (+pre-amplification)	fM (-pre-amplification) aM (+pre-amplification)	0.2-7.9 copies/ μL (-UDG) 15-25 copies/ μL (+UDG)
Assay time	60-120 min, 37°C (RPA+CRISPR)	30 min, 65°C (RT-LAMP) + 10-20 min, 37°C (CRISPR)	30 min, 65°C (RT-LAMP) + 3 min, 37°C (CRISPR)
Readout mode	Single mode: Fluorescence (FL)	Single mode: FL or lateral flow assay (LFA)	Dual mode reporters developed: FL+LFA in the same reaction
Key equipment	FL mode: Heater & plate reader	FL mode: Heater & plate reader LFA mode: Heater	FL mode: Heater & 460 nm LED LFA mode: Heater
Sample type	DNA & anal swabs	DNA, RNA, or heteroduplex (first-reported), & synthetic urine	RNA & nasal swabs
Extraction	Crude DNA extraction	Purified DNA and RNA	DNA extraction buffer at 15 min @ 65°C + 2 min @ 98°C & Maxwell for clinical samples.
Cold chain	Freezer required	Freezer required	ENHANCEv2 components lyophilized. Stable for 30 days at 25°C. RT-LAMP not lyophilized.
Reagent cost per reaction	<\$1 (F)	<\$1 (F), <\$3 (LFA)	<\$1 (F), <\$3 (LFA)
	First study on Cas12a detection. Several applications including COVID-19 diagnostics reported.	First study on enhancing collateral activity of a Cas by engineering crRNAs & enhancing sensitivity.	First study on developing and validating dual mode reporters. Lyophilized, stable for 30 days.

Fig. S7. Comparison of Cas12a-based DETECTR, ENHANCEv1, and ENHANCEv2.

Minor comments:

R2.2. Can the authors speculate further on patient sample p12, which gives negative results on RT-PCR but positive on ENHANCE(v2)? What would be the possible explanations?

Response to R2.2. Patient samples were obtained from the CTSI at Shands Hospital without any patient information and without Ct values. The sample quality was highly variable due to factors not in our control (such as sample collection), and the only information provided were predetermined verdicts of whether the samples were positive or negative. Although determinations were already made on whether SARS-CoV-2 was detected, we have conducted our own tests on Ct values and RT-PCR for robust comparison of the ENHANCEv2 with the standard approved method by the CDC. Patient sample 12 was designated as “positive” but, with repeated testing, displayed Ct values above the threshold designated by the CDC. Therefore, according to our predetermined objective method for comparison, the sample had to be designated as not in agreement. It is possible that the sample was truly negative and that there was carry-over contamination that influenced the sample in our testing or that the ENHANCEv2 test was detecting small amounts of the SARS-CoV-2 N gene that the RT-PCR test was not able to detect at a level that is considered “significantly positive” according to the threshold. As you can see in fig. 3a, there was not a significant increase in the RFU for the E gene for either the wildtype or the variant or for the N gene of the wildtype - it was only noticeably higher for the N gene variant detection. Also, for the lateral-based test, as can be seen in fig. S2, two lines are visible, indicating the background signal is high. This is not the case in most of the other samples. It should also be noted that after extraction, the concentration of total nucleic acid content of that sample and a few others were particularly low, so the sample was not the best quality.

R2.3. Page 6: can the authors provide references to support the claim “correlation between cis-cleavage and trans-cleavage capabilities” of Cas enzymes?

Response to R2.3. We would like to point out to the reviewer that this correlation between cis-cleavage and trans-cleavage capabilities applied to AsCas12a and LbCas12a. We have performed the experiments to show this in our previous publication (Nguyen et al., *Nat Comm.*, 2020) titled “Enhancement of trans-cleavage activity of Cas12a with engineered crRNA enables amplified nucleic acid detection”, specifically, in supplementary figures 11 and 12 of this previous publication, we observed a similar effect of the modified crRNA on cis-cleavage and trans-cleavage ability.

We also would like to point out that, the trans-cleavage activity of Cas12a does not necessarily always commensurate with the cis-cleavage activity. An example of this phenomenon is FnCas12a. While having a robust cis-cleavage activity and is widely used for genome editing, FnCas12a exhibits poor trans-cleavage activity, which was shown in our aforementioned publication (fig. 3e) and other publications such as Fuchs, R *et al.* titled “Cas12a trans-cleavage can be modulated in vitro and is active on ssDNA, dsDNA, and RNA”.

R2.4. The authors mentioned LoD was determined for 6 crRNAs, but only showed results from 4 crRNAs

Response to R2.4. The 6 crRNAs were tested for screening purposes. crRNAs for E1 and R1 did not perform as well as E2 and R2 in terms of LoD. Both N gene crRNAs performed well, so the LoDs were tested for comparison purposes to then narrow it down the crRNA that performed better: N2. Only two SARS-CoV-2 genes were necessary within the diagnostic test, so N, E, and R were further narrowed down to the N and E genes since these performed the best.

R2.5. Page 7: ENHANCEv1 is misspelled

Response to R2.5. Thank you for letting us know about this. It appears that there was an inconsistency between versions of the manuscripts where this was fixed in some and not in others.

R2.6. Page 8: the authors referred to Figure 3F but I do not see this subfig. in Figure 3. Labels A and B were also missing from Figure 3.

Response to R2.6. Thank you for catching this. Figures were rearranged in between versions off the manuscript. This has been updated and should read as fig. 4e.

Reviewer #3 (Remarks to the Author):

R3.1. The work of Nguyen et al provides clinical validation of the CRISPR/Cas12a system for the detection of SARS-CoV2. This work is well-written and timely.

I have the following comments on this manuscript.

1- Although the authors have shown that the modified crRNA exhibits much higher trans cleavage activity, it is not clear what impact this enhancement has on the sensitivity or specificity of the assay. In other words, would this enhancement lead to improvements in the sensitivity/ specificity, or just it may lead to a reduction of the assay time? This point needs to be sufficiently addressed.

Response to R3.1. We thank the reviewer for the great feedback. When we started developing ENHANCEv1, sensitivity and specificity were the two main pillars of our study. In our previous publication titled “Enhancement of trans-cleavage activity of Cas12a with engineered crRNA enables amplified nucleic acid detection” in *Nat Comm*, we detailed how the modified crRNA exhibits enhanced sensitivity across multiple targets tested such as HIV, PCA3, GFP, and SARS-CoV-2 genes. We also observed a slight increase in specificity of ENHANCE compared to the wild-type Cas12a system when normalized to the wild-type. All the details can be found in the publication mentioned above.

R3.2. Since this work aims primarily on the validation of the CRISPR/Cas12 assay. I would expect that the improved system tested on an expanded number of clinical samples with different ct values. I recommend the authors use clinical samples of ct values below and above 30 to explore the concordance of the sensitivity of this assay with the RT-qPCR. Such quantitative data are needed to substantiate the message of the manuscript and reflect that these improvements influence the performance of the assay on the clinical samples. Not just the strength of the signals.

Response to R3.2. We appreciate your feedback. We have updated the manuscript with the addition of the Ct values to fig. 3a, 3b, and 4e in the main text to provide more quantitative information to the reader. As you can see from the updated figures, a wide range of Ct values were included. CDC guidelines use 40 as the threshold for determining whether a sample is positive or negative, so we followed those guidelines for classification.

Additionally, we would like to share with the reviewer some information regarding the patient sample collection process. We collected the samples mostly from the UF CTSI center and Boca Biolistics where only pre-determined results of whether the samples were positive or negative were given to us. We were not able to take control of patient sample quality. On top of that, we performed RT-qPCR in our lab using recommended CDC guidelines to determine the Ct values for all the samples. As a result, this process greatly enriched our validation study as the samples were blinded (described in materials and methods) before extraction and detection assays.

R3.3. I wonder whether the authors have tested their system in a one-pot assay which is of great importance for the development of the POC system. The authors should sufficiently discuss what could be done to limit the cross-contamination.

Response to R3.3. Thank you for the suggestion. We have been working on optimizing conditions using a Cas protein that can function for a one-pot version of ENHANCE as well as using a recently purified polymerase to allow for the RT-LAMP reaction to be lyophilized. Hopefully, that will work out and we can include it in a follow-up publication.

Many steps have been taken in the lab to reduce contamination during this process, including using filtered tips (and always changing tips between uses), cleaning the area with RNase away, and prepping LAMP/RT-LAMP reactions under the hood using pre-portioned reagents. We also incorporate UDG and dUTP in the CRISPR detection reactions. For fluorescence detection, RNaseP was always used as a control for QC failure and negative controls were used to ensure master mixes did not contain contamination.

We have added the following highlighted sentences to the discussion section of the main text:

Similar to SHERLOCK and DETECTR, a major drawback to this technology is the possibility of carryover contamination. CRISPR-based detection system provides two check points for detection: (1) an isothermal pre-amplification step such as RT-LAMP that converts SARS-CoV-2 genomic RNA into millions of dsDNA copies and (2) a CRISPR reaction **in which** crRNA:LbCas12a complex targets the amplified dsDNA. Since this detection platform is not a continuous step, contamination might occur. Therefore, proper reaction setup must be met. This major disadvantage motivated us to develop ENHANCEv2. We also incorporated UDG and dUTP in the CRISPR detection reactions. ENHANCEv2 uses RT-LAMP master mix containing UDG which has been shown to greatly eliminate carryover contamination³⁵. **Cleaning the area with RNase away, using filtered pipette tips, and preparing LAMP/RT-LAMP reactions within a nucleic acid workstation (with HEPA filtration and UV light irradiation) can further reduce contamination.** For fluorescence detection, RNaseP was used as a control for QC failure, and negative controls were used to ensure master mixes did not contain contamination. The lyophilized CRISPR version not only significantly reduces the reaction time but also simplifies the needs for CRISPR reaction setup. The reaction can be reconstituted with the addition of RNase-free water and RT-LAMP products. The method further promotes large-scale manufacturing, eases the challenge of reagent storage and handling, and would enable easier deployment across the globe. **Additional lyophilization of the RT-LAMP reaction may reduce contamination and allow for a more practical at-home testing solution.**

R3.4. I think the authors need to modify the title to reflect the message of the manuscript which is the clinical validation of the system. More quantitative data are

required with comparisons with RT-qPCR with different Ct values.

Response to R3.4. This is a great suggestion. We have modified our title to fit the clinical validation aspect of the study. It now reads:

Clinical Validation of Engineered CRISPR/Cas12a For Rapid SARS-CoV-2 Detection

R3.5. Authors need to cite other works on the CRISPR/Cas12 system including “iSCAN: An RT-LAMP-coupled CRISPR-Cas12 module for rapid, sensitive detection of SARS-CoV-2” by Ali et al 2020.

Response to R3.5. Thank you for the recommendation. We think that this paper is a great study; therefore, we have updated our reference list that includes this publication.

R3.6. Authors need to provide one paragraph as an update on the status of CRISPR technologies for SARS-CoV-2 detection and current challenges and context and the importance of this work.

Response to R3.6. We thank the reviewer for the valuable suggestion. We have added a paragraph in our main text detailing the status of CRISPR technologies, their advantages and disadvantages, and how our work contributes to the CRISPR diagnostics field in general. The paragraph read as follows:

The future of CRISPR-based diagnostic platforms is bright. In addition to SHERLOCK and DETECTR technologies receiving emergency use authorization by the FDA, recent collaboration of MilliporeSigma and Mammoth Biosciences brings forward scale-up of a new SARS-CoV-2 test, allowing for rapid turnaround time and massive testing capability³⁶. Isothermal detection using CRISPR enables rapid results without the need for expensive equipment. However, since many CRISPR detection methods involve a highly sensitive pre-amplification, contamination remains a challenge. This problem is not specific to CRISPR tests as the gold standard RT-qPCR must also overcome this issue. We hope that the improved capability and stability of ENHANCEv2 will aid in global deployment towards curbing the COVID-19 pandemic.

Minor

R3.7. 1- Please fix “ we developed a lyophilized a version of ENHANCEv2 system” to “ we developed a lyophilized version of ENHANCEv2 system”

3- Please fix “ The ENHANCE system works robustly with a wide range of a magnesium concentration (3mM- 13mM)” to “ The ENHANCE system works robustly with a wide range of magnesium concentrations (3mM- 13mM)”

Response to R3.7. We thank the reviewer for catching these mistakes. We have fixed them and updated in our manuscript.

Reviewer #4 (Remarks to the Author):

R4.1. It was an interesting work that the authors introduce ENHANCE system (V1 and V2), both showed a significant increase in sensitivity while maintaining specificity even at low magnesium concentrations. ENHANCEv2 system, which utilizes modified crRNAs from ENHANCE, a mutated LbCas12a protein (Cas12aD156R) 22-25 to further amplify the signal, and a dual reporter construct that allows each sample to be read using a fluorescence-based and a lateral flow assay format in the same reaction. The lyophilized CRISPR reaction in ENHANCEv2 occurs at an accelerated rate and preserves for a long period of time upon room temperature storage. The manuscript is well organized and easy to follow, but I still have some comments as follows:

1st, As the authors state the limit of detection (LoD) the ENHANCE platform (crN2) is 0.2 copies/ μ L, while the LoD of the US FDA EUA-approved CDC assay for detection of SARS-CoV-2 is 1 copies/ μ L. However, ENHANCE platform add 5 μ L template to reaction, and CDC assay add 2 μ L template to reaction. Actually, both methods have same sensitivity, about 1~2 copy/rxn. LoD of the ENHANCE platform described in this manuscript will confused the readers that ENHANCE has higher sensitivity than CDC assay.

Response to R4.1. We appreciate the reviewer's feedback regarding the sensitivity of the ENHANCE platform as well as the volume of sample input in the detection reaction. When designing the validation experiments, we consulted with the CDC guidelines and followed them as strictly as possible. We are not sure which CDC assay the reviewer was referring to. Many of the protocols we consulted use 5 μ L of extracted nucleic acid samples in the amplification reaction whether it is RT-qPCR or RT-LAMP. For instance, the RT-qPCR recipe on page 19 of the CDC 2019-nCoV panel at <https://www.fda.gov/media/134922/download> uses 5 μ L of nucleic acid sample for a wide variety of commercially available and CDC recommended RT-qPCR master mixes. Another example would be the protocol for rapid detection of SARS-CoV-2 using DETECTR developed by Mammoth Biosciences at https://mammoth.bio/wp-content/uploads/2020/04/200423-A-protocol-for-rapid-detection-of-SARS-CoV-2-using-CRISPR-diagnostics_3.pdf, which also uses 5 μ L of patient samples.

While preparing the manuscript, we always made sure we do not confuse the readers about the sensitivity of the CRISPR detection platform. In fact, RT-qPCR is considered as a gold standard nucleic acid detection platform because of its single copy detection capability. Researchers in the CRISPR diagnostics field are looking at RT-qPCR as a standard for improving the CRISPR limit of detection. We did not claim that the ENHANCE achieved higher sensitivity than Rt-qPCR. Therefore, we have commented in our manuscript as follows:

"These results demonstrate that the CRISPR-ENHANCE achieved a comparable sensitivity to traditional RT-qPCR testing of SARS-CoV-2 in less than an hour without sophisticated equipment".

R4.2. 2nd, in Discussion section. Authors discussed three samples of conflicting results. Do you know what is the pre-determined method? QPCR or clinical diagnosis? QPCR should not be the only diagnostic criteria. Authors should be make sure the sample type, not only based on QPCR. Patient sample p01 was pre-determined as negative upon arrival in your lab, it is also indicated as negative with your ENHANCE assay, but revalidation with RT-qPCR assay indicated it as a positive sample. Since ENHANCE has same sensitivity with QPCR, why you concluded that this sample was a false negative based on your analysis approach, whether it is possible that QPCR is wrong? The patient sample p12 was predetermined to be a negative sample and was detected negative with the qPCR, however, it was consistently detected as positive by ENHANCE v1 and v2 methods. Is it possible that ENHANCE is right and p12 is false negative?

Response to R4.2. We appreciate your feedback. This is a great point, and we agree with the reviewer on this. Since samples were obtained from the CTSI at Shands Hospital without any information on the methods used for detection, without Ct Values, and without any clinical information about the patients, such as symptoms, RT-qPCR was used as the basis for comparison for ENHANCEv1 and ENHANCEv2 on each sample in order to have a quantitative idea of the Ct values. Many of the samples were not of the best quality and had low concentration values after extraction, possibly due to collection methods which were outside of our control. Since there was a very limited amount of each sample available for testing, it was not possible to compare it to other methods, especially since NGS is the only option beside RT-qPCR. Particularly, we addressed patient 12 in more detail in the second paragraph of our response to reviewer 2 if you would like more information. However, in short, yes, it is possible that ENHANCE is detecting small concentrations of the SARS-CoV-2 N gene that the RT-qPCR was not detecting, especially due to the low concentration of the sample. In fact, the clinical diagnosis of that sample was that it was positive, not negative, which matched the results from ENHANCE. However, unfortunately, since we do not have any detailed information on the clinical diagnosis of the sample other than a simple positive/negative determination, it would not be scientifically objective to use their determination since we do not have any results that can demonstrate that. Thus, we determined that it was best practice to use the results that we personally obtained for comparison.

R4.3. 3rd, In Main section, the third paragraph. Authors mentioned that Cas12a-based DETECTR has a limit of detection of around 20 copies/ μ L, while SHERLOCK is based on Cas13a and is timeconsuming. It is wrong. I read the paper "CRISPR-Cas12-based detection of SARS-CoV-2", the LoD of DETECTR is 10 copies/ μ L, and 2 μ L template add to the reaction, 20 copy/rxn. Please make sure the LoD of DETECTR and SHERLOCK.

Response to R4.3. We thank the reviewer for pointing this out. We apologize for citing the wrong source.

We based the limit of detection on the FDA EUA approval on clinical validation of both SHERLOCK and DETECTR technologies. The limit of detection was defined as the lowest concentration in clinically relevant samples where at least 19/20 replicates were detected positive, resulting in 95% confidence. The reviewer can refer to the two FDA EUA documents below:

<https://www.fda.gov/media/139937/download>

<https://www.fda.gov/media/137746/download>

In table 4 on page 5 of the first document regarding the DETECTR technology, the limit of detection was concluded as 20 copies/ μ L corresponding to 100 copies/reaction (5 μ L of nucleic acid sample).

In table 15A on page 24 of the second document regarding the SHERLOCK technology, the limit of detection was concluded as 6.75 μ L.

We believe that the LoD of DETECTR described in Broughton et al. was for synthetic nucleic acid target fragments of SARS-CoV-2 genes, and not clinically relevant samples. We again apologize for citing the wrong source; therefore, we have updated the reference list to reflect the correct information.

R4.4. 4th, In Main section, the third paragraph, authors stated that SHERLOCK utilizing Recombinase Polymerase Amplification (RPA) and has a limit of detection of around 6.75 copies/ μ L, while DETECTR utilizing Reverse Transcription Loop Mediated Isothermal Amplification (RT-LAMP), has a limit of detection of around 20 copies/ μ L. But the last sentence of this paragraph, author write “The RT-LAMP, performed at a constant temperature of 60-65°C, is significantly less timeconsuming due to its “cauliflower-like” logistic growth amplification pattern. It is also more sensitive than RPA and can be easily adopted without supply-chain issues“. It is could be confused.

Response to R4.4. We appreciate the reviewer’s feedback regarding the wording of our manuscript. We have updated our manuscript as follows:

The RT-LAMP, performed at a constant temperature of 60-65°C, is significantly less time-consuming due to the incorporation of loop primers that facilitate subsequent primer binding and amplification at multiple sites.

R4.5. 5th, ENHANCEv2 uses RT-LAMP master mix containing UDG which has been shown to greatly eliminate carryover contamination, this paper should be cited. Analytical Chemistry 2019 91 (17), 11362-11366

Response to R4.5. We thank the reviewer for the genuine feedback. We have updated our reference list to include the specified publication.

Reviewers' comments:

Reviewer #1 (Remarks to the Author):

The authors have addressed most, but not all, of my concerns.

I do still feel strongly that the authors should demonstrate that the LAMP reagents can be lyophilized and used in ENHANCEv2.

If ENHANCEv2 is not compatible with lyophilized LAMP reagents, then it will not be easier to store/handle/deploy the assay across the globe. Lyophilization of only part of the assay is a step in the right direction, but it is not sufficient to actually use the assay at a site without access to cold storage.

Reviewer #2 (Remarks to the Author):

The authors have adequately addressed my and other reviewers' concerns. The reporting of Ct values of clinical samples (and the manuscript title change) is fitting for the clinical validation nature of this study, though I am surprised that the Ct values do not correlate with the fluorescence intensity values from either version of ENHANCE (and it is not because the fluorescence signal is saturating). It would be good for the authors to comment on this in the further revised manuscript.

Update: I was also asked to specifically look at the authors' response to comments from Reviewer 3, who was not available to look at the revised manuscript.

The authors satisfactorily addressed all concerns raised by this reviewer.

Reviewer #3 (Remarks to the Author):

I agree to accept this manuscript.

We appreciate the reviewers' feedback on our manuscript titled "**Engineered CRISPR/Cas12a Enables Rapid SARS-CoV-2 Detection**". We have addressed all the comments below with responses marked in blue in this **Revision 2** and **changes to the manuscript highlighted in yellow**.

Reviewer #1 (Remarks to the Author):

The authors have addressed most, but not all, of my concerns.

I do still feel strongly that the authors should demonstrate that the LAMP reagents can be lyophilized and used in ENHANCEv2.

If ENHANCEv2 is not compatible with lyophilized LAMP reagents, then it will not be easier to store/handle/deploy the assay across the globe. Lyophilization of only part of the assay is a step in the right direction, but it is not sufficient to actually use the assay at a site without access to cold storage.

Response: We once again appreciate this important feedback from the reviewer. Because the commercial RT-LAMP kits (NEB) contain glycerol and are proprietary, they cannot be directly lyophilized or reformulated using commercial available proteins. Therefore, as indicated in the response to the comments last time (Reviewer#1, comment# 3), we have successfully purified necessary proteins for RT-LAMP reaction including Bst polymerase and reverse transcriptase RTx(exo-) in-house. These proteins are considered alternatives to the Warmstart® Master Mix from New England Biolabs. We are delighted to tell the reviewer that we have successfully lyophilized our RT-LAMP reagents, and they worked well. We observed 1-2 minutes slower in amplification compared to the engineered Bst polymerase 2.0 and reverse transcriptase RTx in the Warmstart® master mix, but the overall performance of our lyophilized reagents complements well with the CRISPR detection reaction and more significantly cost-effective.

We have updated the Materials and Methods section in our manuscript to reflect the addition of Bst polymerase, reverse transcriptase (RTx-exo-) expression and purification, and lyophilized steps of RT-LAMP reagents. We have also added Fig. S7 to the Supplementary Information section with data showing the RT-LAMP reaction using our lyophilized LAMP reagents compared to Warmstart® Master Mix. We hope that with this complete lyophilization pipeline from RT-LAMP to CRISPR reagents, we will be able to help with the easy distribution, handling and deployment of the assays in remote areas.

Please see below the changes to the manuscript:

Protein expression and purification

Bst-LF polymerase expression plasmid obtained as a gift from Drew Endy & Philippa Marrack (Addgene plasmid # 153313). Br512 (an engineered version of Bst polymerase) and reverse transcriptase RTx (exo-) were obtained as a gift from Andrew Ellington (Addgene plasmid # 161875 and # 145028, respectively). Bst-LF polymerase and Br512 polymerase were expressed and purified following *Maranhao A et al.*, and RTx(exo-) was expressed and purified following *Bhadra S et al.*

Lyophilization of RT-LAMP reagents

One reaction of the RT-LAMP assay reagent mixture was prepared by combining 35 nanomoles dNTPs, 2.5 μ L of 10X LAMP primer mix, 25 picomoles of Br512 (or Bst-LF) polymerase, 0.1 μ g of RTx(exo-), and 1.25 μ moles of D-(+)-trehalose, anhydrous. The mixture was frozen for 1 hour at -80°C prior to freeze-drying using the Labconco lyophilizer for 24 hours.

The lyophilized reaction mixture was reconstituted with 1X isothermal buffer (20 mM Tris-HCl, 10 mM $(\text{NH}_4)_2\text{SO}_4$, 50 mM KCl, 10 mM MgSO_4 , PH = 8.8 at 25°C). The RT-LAMP reaction was then readily initiated by adding RNA samples.

Fig. S7. Lyophilization and functional testing of RT-LAMP reactions. (a) Representation of lyophilized RT-LAMP reactions using Br512 polymerase and reverse transcriptase RTx(exo). (b) Representation of lyophilized RT-LAMP reactions using Bst-LF polymerase and reverse transcriptase RTx(exo). (c) RT-LAMP amplification of lyophilized reagents compared to commercial Warmstart® Master mix (New England Biolabs). SYTO9 dye was used to track amplification for each replicate (n = 2 biological replicates). As seen above, Bst-LF + RTx performs as robustly as the Warmstart® Master mix, whereas the Br512 + RTx combination produces non-specific signal the earliest. (d) CRISPR detection reaction of amplified target using the lyophilized RT-LAMP reagents from (a) and (b) compared to the commercial Warmstart® Master mix (New England Biolabs). The fluorescence intensities were taken at t = 30 minutes, with n = 4 biological replicates.

Reviewer #2 (Remarks to the Author):

The authors have adequately addressed my and other reviewers' concerns. The reporting of Ct values of clinical samples (and the manuscript title change) is fitting for the clinical validation nature of this study, though I am surprised that the Ct values do not correlate with the fluorescence intensity values from either version of ENHANCE (and it is not because the fluorescence signal is saturating). It would be good for the authors to comment on this in the further revised manuscript.

We thank you the reviewer for the great comment. Based off of our previous observations, Ct values do not always correlate well with the fluorescence intensity from the CRISPR reaction. There are multiple factors that could affect this disproportionality between RT-LAMP reaction and the CRISPR reaction, such as the followings:

1. RT-LAMP reactions are fast reactions. Typically, the amplicons are produced as fast as 7-10 minutes (Refer Fig. S7). However, in order to be consistent in detecting patient samples, we applied a 30-minute isothermal incubation to all the samples. Depending on the viral load in each sample, saturation in products being amplified can vary, but 30 minutes does allow some small viral load samples (high Ct values) to catch up to the saturation point along with low Ct value samples. Therefore, Ct values sometimes do not correlate with CRISPR fluorescence intensity.
2. We have observed that too much amplified product from the RT-LAMP reaction can have an inhibitory effect on the CRISPR detection reaction. This observation corroborates a study by Li Z et al. titled "A Chemical-Enhanced System for CRISPR-Based Nucleic Acid Detection" in which they have noted a similar phenomenon.
3. In addition, as the copy number of nucleic acid template decreases, there is an increasing variation in RT-LAMP reactions in terms of time to reach signification amplicons, as demonstrated by Hardinge P et al. titled "Full Dynamic Range Quantification using Loop-mediated Amplification (LAMP) by Combining Analysis of Amplification Timing and Variance between Replicates at Low Copy Number". This phenomenon was also observed in SHERLOCK, DETECTR, and StopCovid technologies.

With that being said, we have added some comments in our revised manuscript. It is as follows:

Based on the results from ENHANCE, we observed that Ct values do not always correlate well with the fluorescence intensity from the CRISPR reaction (figure 3d). It is possibly due to an inhibitory effect on the CRISPR detection reaction by excessive amplified product. This observation corroborates a study by Li Z et al. in which they have noted a similar phenomenon. In addition, as the copy number of nucleic acid template decreases, there is an increasing variation in RT-LAMP reactions in terms of time to reach signification amplicons, as demonstrated by Hardinge P et al.

REVIEWERS' COMMENTS:

Reviewer #2 (Remarks to the Author):

The authors have addressed the reviewers' concerns in full; no further revision is necessary.

Update:

I was further asked to specifically comment on the authors' response to comments from Reviewer 1.

The authors have satisfactorily addressed Reviewer 1's concerns. The additional work on generating RTx and BST-LF proteins and using them in lyophilized LAMP reactions clearly demonstrated the field-deployable nature of the authors' method. This additional work really strengthened an already strong manuscript in my opinion. No further revision is requested.